# Highly regulated, diversifying NTP-dependent biological conflict systems with implications for the emergence of multicellularity

**Gurmeet Kaur†, A Maxwell Burroughs†, Lakshminarayan M Iyer, L Aravind\***

Computational Biology Branch, National Center for Biotechnology Information, National Library of Medicine, National Institutes of Health, Bethesda, United States

**Abstract** Social cellular aggregation or multicellular organization pose increased risk of transmission of infections through the system upon infection of a single cell. The generality of the evolutionary responses to this outside of Metazoa remains unclear. We report the discovery of several thematically unified, remarkable biological conflict systems preponderantly present in multicellular prokaryotes. These combine thresholding mechanisms utilizing NTPase chaperones (the MoxR-vWA couple), GTPases and proteolytic cascades with hypervariable effectors, which vary either by using a reverse transcriptase-dependent diversity-generating system or through a system of acquisition of diverse protein modules, typically in inactive form, from various cellular subsystems. Conciliant lines of evidence indicate their deployment against invasive entities, like viruses, to limit their spread in multicellular/social contexts via physical containment, dominant-negative interactions or apoptosis. These findings argue for both a similar operational 'grammar' and shared protein domains in the sensing and limiting of infections during the multiple emergences of multicellularity.

**\*For correspondence:**
aravind@ncbi.nlm.nih.gov

†These authors contributed equally to this work

**Competing interests:** The authors declare that no competing interests exist.

## Introduction

Genetic systems are locked in multilevel conflicts which span all levels of biological organization. These include intra-genomic conflicts between genetic elements within genomes, the conflict between distinct replicons in a cell and inter-organismal conflicts of unicellular and multicellular organisms between members of same or different species (*Aravind et al., 2012*; *Dawkins and Krebs, 1979*; *Hurst et al., 1996*; *Werren, 2011*; *Smith and Price, 1973*; *Austin et al., 2009*). As success in these biological conflicts is central to the survival of organisms, the genetically encoded traits associated with them are under intense selection. As a result, these adaptations are part of a continuing arms-race between the genetic entities locked in conflict, wherein each tries to outperform the others resulting in constant selection for better armaments (*Dawkins and Krebs, 1979*; *Thompson, 1994*; *Iyer et al., 2011a*; *Van Melderen, 2010*). Consequently, biological conflicts have left their prominent marks on the physiological, morphological and behavioral adaptations of organisms and show up as clear-cut signatures in the genomes of organisms. One of the recent triumphs of comparative genomics has been the successful prediction of genomic signatures of biological conflict and counter-conflict systems. Consequently, these genomic imprints of biological conflicts have served as excellent models to study evolution on a 'fast-track' (*Aravind et al., 2012*). Additionally, they have also provided 'work-horse' reagents for molecular biology such as restriction enzymes, nucleic-acid-modifying enzymes and CRISPR/Cas9 technologies (*Murray, 2000*; *Lavender et al., 2018*; *Gaj et al., 2013*).

**eLife digest** Bacteria are the most numerous lifeforms on the planet. Most bacteria live as single cells that grow and multiply independently within larger communities of microbes. However, some bacterial cells assemble to form more complex structures where individual cells might perform distinct roles. Such bacteria are referred to as 'multicellular bacteria'. For example, cells of bacteria known as *Streptomyces* collectively form filaments that help the bacteria collect nutrients from their food sources, and aerial structures bearing reproductive spores.

Bacteria in these filaments may come into contact with many other microbes in their surroundings including other bacteria within the same filament, other species of bacteria, and viruses. These contacts often lead to conflict, for example, if the microbes compete with each other for nutrients or if a virus tries to attack the bacteria.

Bacteria have evolved immune systems that detect other microbes and use antibiotics, toxins and other defense mechanisms against them. Compared to single-celled bacteria, multicellular bacteria may be more vulnerable to threats from viruses because once a virus has overcome the defenses of one cell in the multicellular assembly, it may be easier for it to kill, or spread to the other cells. However, it is not clear how these systems evolved to deal with the unique problems of multicellular bacteria. Now, Kaur, Burroughs et al. have used computational approaches to search for new immune systems in diverse multicellular bacteria. The new classes of systems they found are each made of different molecular components, but all require a large input of energy to be activated. This activation barrier prevents the bacterial cells from deploying weapons unless the signal from the enemy microbe crosses a high enough threshold.

Many tools used in molecular biology, and increasingly in medicine, have been derived from the immune systems of bacteria, such as the enzymes that cut or edit DNA. The findings of Kaur, Burroughs et al. may aid the development of new tools that specifically bind to viruses or other dangerous microbes, or inhibit their ability to interact with components in cells. The next step would be to perform experiments using some of the immune systems identified in this work.

Much of the molecular weaponry in these biological conflicts takes the form of effectors that accomplish their action via attacks on the biomolecules that transmit information in 'the central dogma', namely genomic DNA, the various RNAs involved in protein synthesis, the protein-synthesizing engine, that is the ribosome, or proteins of the replication, transcription and translation systems (*Werren, 2011*; *Smith and Price, 1973*; *Iyer et al., 2011a*; *Leplae et al., 2011*; *Proft, 2005*; *Walsh, 2003*). Less-frequent modes of action include attacks on components of the signal-transduction systems or direct rupture of cells by membrane perforation or destabilization (*Rappuoli and Montecucco, 1997*; *Gilbert, 2002*; *Burroughs and Aravind, 2016*; *Burroughs et al., 2015*; *Lambert, 1978*). In terms of their chemistry, the weaponry deployed in these conflicts takes a very diverse form ranging from low-molecular-weight 'toxins' and 'antibiotics' synthesized by the products of a range of multi-gene biosynthetic operons to proteinaceous toxins that feature some of the largest-described polypeptides in their ranks (*Walsh, 2003*; *Iyer et al., 2017*). A consequence of the intense natural selection acting on these molecules is their rapid evolution with extreme diversity in terms of structure and sequence. This often serves as a telltale marker to identify them through sequence-analysis and comparative genomics (*Dawkins and Krebs, 1979*; *Thompson, 1994*; *Iyer et al., 2011a*; *Van Melderen, 2010*).

Effector-deployment poses several challenges to the host. First and foremost, given that the effectors frequently target the central dogma systems, which tend to be universally shared, there is the need for mechanisms to specifically target rival or non-self genetic systems as opposed to self biomolecules. This is most commonly achieved by specific antitoxins (e.g. in toxin-antitoxin (TA) systems) and immunity proteins (e.g. in polymorphic toxin and related systems) that neutralize the effector when not required (*Leplae et al., 2011*; *Anantharaman and Aravind, 2003a*; *Kobayashi, 2001*; *Daw and Falkiner, 1996*; *Zhang et al., 2012*; *Zhang et al., 2011*). However, this problem is even more acute in cases where the biological conflict is between genomes in the same cell, such as those between the host genome and invasive nucleic acids like viruses and plasmids. In the simplest cases, the solution takes the form of a mechanism to distinguish self vs. non-self, with the most

well-known form being the 'marking' of self DNA by modification of the bases or the backbone by enzymes, which are coupled to restriction enzyme effectors that target nucleic acids based on the differences in their modification status. This is the basic principle behind the diverse restriction-modification (R-M) systems (*Leplae et al., 2011*; *Anantharaman and Aravind, 2003a*; *Kobayashi, 2001*; *Daw and Falkiner, 1996*). Another mechanism, typical of the CRISPR/Cas and Piwi-dependent systems, is the direct detection of non-self nucleic acids through complementary pairing mediated by guide nucleic acids (*O'Connell et al., 2014*; *Ameres et al., 2007*).

However, these conflicts with invasive nucleic acids often present more complex decision points. For instance, a cell might sacrifice itself via apoptosis or cell suicide if it can thereby prevent the genetic furtherance of the antagonistic or infectious entity replicating within it (*Makarova et al., 2012*). While this is fitness-nullifying for the cell, the behavior might still be genetically profitable if one were to consider included fitness accrued via kin benefiting from the suicidal act (*Bourke, 2014*; *Hamilton, 1964*). However, such a suicidal action, wherein the effectors are deployed against self molecules, needs to be tightly regulated to prevent its inappropriate activation when not required. Moreover, effector-production and -deployment themselves divert resources from processes central to actual organismal proliferation. Hence, the initiation of effector actions leading to dormancy or even suicide need to be weighed against alternative house-keeping functions leading to proliferation. Given these stakes, it is becoming increasingly clear that the effector deployment step in conflict systems is often subject to threshold detection mechanisms (*Burroughs et al., 2015*) that limit the serious energetic or even existential consequences of unintentional effector deployment.

Our recent studies and follow-up wet-lab confirmations have led to the uncovering of major thresholding mechanisms that control key biological conflict systems deployed by cells against invasive nucleic acids such as viruses (*Burroughs et al., 2015*; *Severin et al., 2018*; *Whiteley et al., 2019*; *Makarova et al., 2014*). Chiefly, these include the use of a diverse array of linear and cyclic nucleotides in a subset of the CRISPR/Cas and the SMODS systems as control mechanisms regulating deployment of the effectors of this system. Briefly, detection of the non-self or invasive entity activates one of multiple evolutionarily unrelated nucleotidyltransferases that then produce a nucleotide signal. It is the detection of this nucleotide by a sensor domain that results in actual effector-deployment (*Burroughs et al., 2015*; *Severin et al., 2018*; *Whiteley et al., 2019*). Thus, the additional step of nucleotide production serves to set a threshold for the deployment of potentially dangerous effectors.

Our studies had also revealed further complexities to the thresholding mechanism in the form of the deployment of chaperones of the AAA+ (ATPases associated with diverse cellular activities) ATPase superfamily of P-loop NTPases. These ATPases channel energy generated by ATP hydrolysis to drive conformational changes (*Neuwald et al., 1999*; *Lupas and Martin, 2002*; *Hanson and Whiteheart, 2005*). Consequently, they have chaperone activity which helps in the (dis)assembly of protein complexes involved in the effector response. Thus, they can act as thresholding 'switches', which regulate the conformation or state of assembly of the effector complexes. We identified such a mechanism as a further regulatory element of certain nucleotide-dependent conflict systems, which has subsequently been confirmed in some wet-lab studies (*Ye et al., 2019*; *Lau et al., 2020*). Here, the TRIP13/Pch2 AAA+ ATPase along with the peptide-binding co-chaperone HORMA domains (*Aravind and Koonin, 1998a*) facilitates conformational switching to set a threshold for response to invasive DNA (*Rappuoli and Montecucco, 1997*; *Burroughs et al., 2015*; *Wojtasz et al., 2009*). More generally, we also observed other examples of the use of chaperones in thresholding mechanisms: in the 3-gene toxin TA system, in addition to the toxin ADP-ribosyltransferase (ART), an antitoxin which is an intrinsically disordered protein, there is the third component, the chaperone SecB that regulates the folding of the antitoxin and affects TA interactions and activity (*Aravind et al., 2015*; *Sala et al., 2013*). The classical NtrC-like AAA+ ATPases which regulate transcription might also be seen as part of a threshold-setting control mechanism (*Banerjee et al., 2019*; *Hsieh et al., 2018*).

Our preliminary investigations suggested that such mechanisms might be more widespread than have been appreciated. Hence, we developed a systematic procedure to identify novel chaperone-based systems and their analogs that regulate effector activity in biological conflicts via thresholding. Consequently, we identified multiple systems, unifiable by several shared mechanistic features, which we group into three categories: (1) chaperone/co-chaperone-based systems centered on conformational changes driven by a MoxR-like AAA+ ATPase and its co-chaperone, the peptide-binding von

Willebrand factor A (vWA) domains (*Snider and Houry, 2006*; *Wong and Houry, 2012*); (2) analogous systems which use GTPases (in some cases with further HSP70 and tubulin family proteins) instead of the MoxR-like AAA+ ATPase (*Leipe et al., 2002*; *Nogales et al., 1998*); (3) systems which share effectors with the first two but have distinct thresholding mechanisms, such as those dependent on proteolytic cascades. Across these systems, we observe associations to further conflict systems which can act within their framework of or in parallel with these systems, including association with a diversity-generating retroelement (DGR) (*Wu et al., 2018*), prokaryotic Ubiquitin-like (UBL) conjugation systems (*Burroughs et al., 2011*; *Iyer et al., 2006*), and a novel class of adaptor domains analogous to the Death-like domains of animal-apoptosis systems. Importantly, we show that these conflict systems are predominantly found in bacteria with complex development and multicellular organization suggesting a link between these features and the strongly regulated effector responses to invasive entities. This is also paralleled in multicellular eukaryotes and helps uncover the general evolutionary principles in the interplay between immunity and developmental complexity.

## Results and discussion

### Systematic identification of conflict systems with novel thresholding mechanisms

To systematically identify novel conflict systems with chaperone-based threshold-setting regulatory mechanisms or their analogs, we drew on the extensive, well-annotated collection of protein domains typical of biological conflict systems. This set includes an array of over 250 effector domains such as diverse families of RNases, DNases, protein/nucleic-acid-modifying enzymes and pore-forming toxins. This collection was assembled from previous systematic analyses of biological conflict systems, such as polymorphic toxins, CR-toxins, TA, R-M, CRISPR/Cas, and nucleotide/small-molecule-activated effectors from previous studies performed by us and others over the past two decades (*Aravind et al., 2012*; *Burroughs et al., 2015*; *Iyer et al., 2017*; *Zhang et al., 2012*; *Aravind et al., 2015*; *Zhang et al., 2016*; *Makarova et al., 2019*). As has been previously noted, such effector domains are often shared across distinct conflict systems and thus served as potential markers for the detection of new conflict systems (*Figure 1A*; *Aravind et al., 2012*; *Burroughs et al., 2015*; *Iyer et al., 2017*; *Zhang et al., 2012*; *Aravind et al., 2015*; *Zhang et al., 2016*; *Makarova et al., 2019*). We used profiles of these domains as seeds in PSI-BLAST searches (typically run up to five iterations) run against a curated collection of 7423 complete genomes (coding for a total of 21646808 proteins) to obtain an initial collection of proteins potentially involved in biological conflicts. Further, selected searches were run against the NCBI *nr* database (June 21$^{st}$, 2019) to obtain a more complete coverage of rarer components and survey the frequency of such systems among currently deposited genomic data more generally.

Proteins in genuine conflict systems typically show: (1) high architectural variability with repeated displacement of effectors within an otherwise well-preserved domain architectural or operonic context (*Aravind et al., 2012*; *Iyer et al., 2011a*; *Van Melderen, 2010*; *Leplae et al., 2011*; *Burroughs et al., 2015*); (2) certain parts of these molecules, usually those encompassing the effector segments that directly interact with the rival entity tend to show high variability which can be measured using Shannon entropy of the sequence alignments. These above features can be observed even between closely related organisms (*Zhang et al., 2012*; *Zhang et al., 2016*; *Krishnan et al., 2018*); (3) Show a high degree of lateral transfer, gene loss and a tendency for extensive difference in presence and absence between closely related organisms (*Van Melderen, 2010*; *Leplae et al., 2011*; *Zhang et al., 2012*; *Iyer et al., 2014*). We leveraged this knowledge to cull a subset of promising candidates displaying these features from the initial collection (*Figure 1A*). We then extended the previous searches to more extensive databases as appropriate (see Materials and methods). We extracted gene-neighborhood information for these candidates and queried the proteins coded by conserved associated genes for chaperone domains using a panel of PSI-BLAST position-specific score matrices (PSSMs, also called sequence profiles) for different classes of chaperones. We also followed these up with parallel searches using hidden Markov models instead of PSSMs with JACKHMMER and HMMSEARCH programs from the HMMER3 package (*Figure 1A*) to detect domains which might have eluded the first approach.

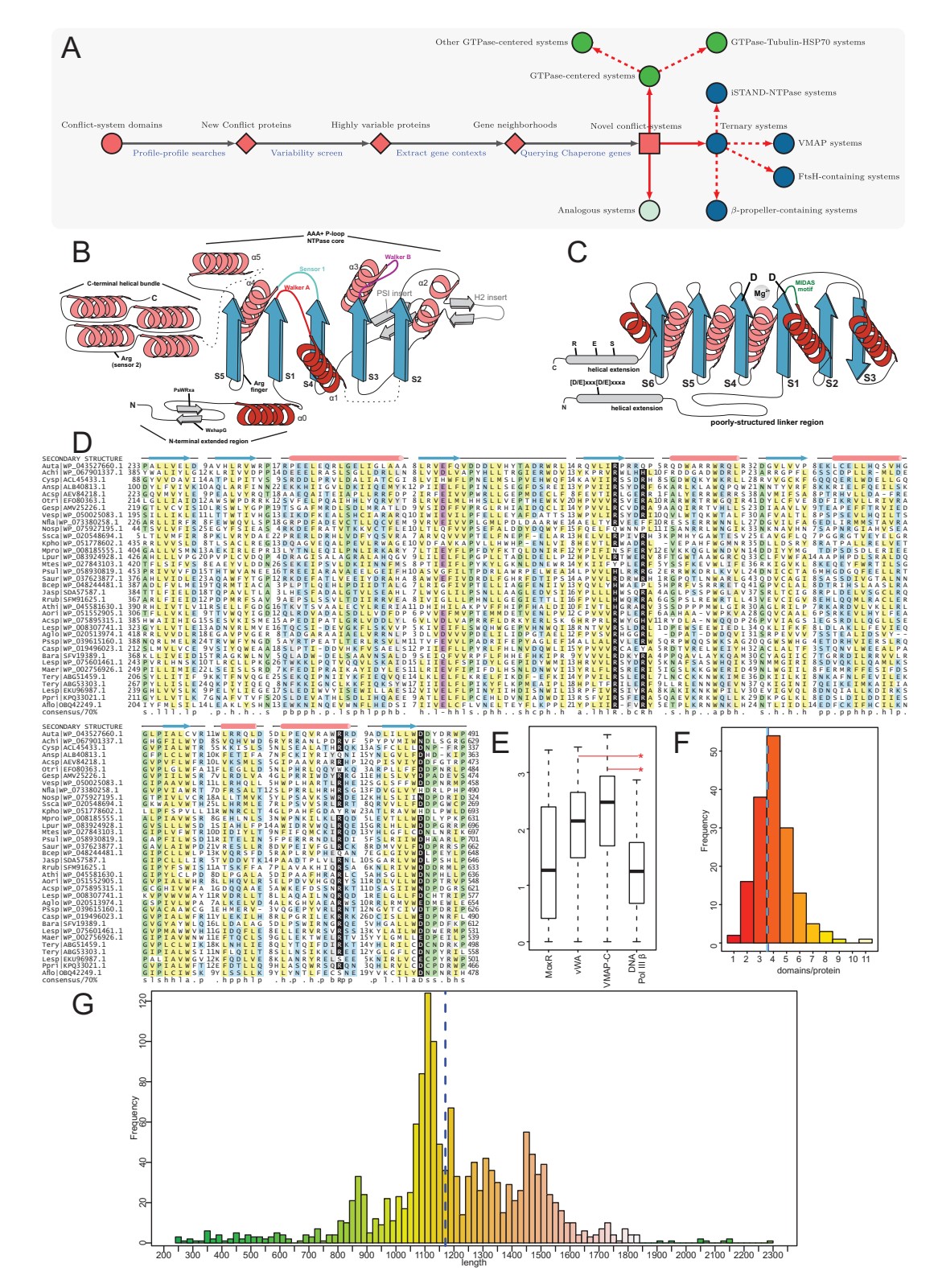

**Figure 1.** Identification of novel conflict systems and overview of core ternary system components. (A) Flowchart showing the process used to identify the conflict systems in this study. (B–C) Topology diagrams of the vWA (B) and MoxR (C) domains highlighting characteristic conserved features. (D) Multiple sequence alignment of VMAP-C domain. Sequences are labeled to the left by organism abbreviation (see *Figure 1—source data 1*) and accession number. Predicted secondary structure is shown on top and the consensus conservation used for coloring is given below. (E) Boxplot

*Figure 1 continued on next page*

*Figure 1 continued*

comparing sequence entropy values for VMAP core ternary system domains, collected from a common set of genomes containing all four components. Significant p-values determined by Wilcoxon Rank-Sum Test: *, p-value<2.2e-16. (**F**) Distribution of the number of effector domains C-terminally fused to the core vWA domain of the classical ternary systems. Dashed vertical lines, colored in cyan and blue, indicate median and mean domain number, respectively. (**G**) Histogram of the length distribution of the core vWA domain-containing component of the VMAP ternary systems, with dashed line as in (**F**).

The online version of this article includes the following source data and figure supplement(s) for figure 1:

**Source data 1.** Comparative genomics data.
**Figure supplement 1.** Histograms of vWA-fused effector protein length distributions by effector domain type.
**Figure supplement 2.** Histograms of vWA-fused effector protein length distributions by taxonomy.

By this procedure we identified four distinct prokaryotic systems with a gene-pair respectively encoding a MoxR-type AAA+ ATPase and its co-chaperone the vWA domain, along with at least one other linked gene coding for a further component of these systems. Hence, we termed all these 'the ternary systems' as they included at least three core components (*Figure 1A*). In these systems the variable effector domain(s) were either fused directly to the vWA domain or encoded as further components of the system. Further, we found another class of systems which combined genes coding for one or two paralogous GTPases of the TRAFAC clade with tubulin homologs and a HSP70 family chaperone protein (*Figure 1A*). Finally, having identified these systems, we also searched for analogous and related systems by investigating if any of the core components are displaced by alternative components or have been lost (*Figure 1A*). This led us to identify a more extensive set of GTPase-centered systems wherein the tubulin and HSP70 components had been lost. In addition, this procedure also allowed us to identify analogous systems wherein the chaperone components had been displaced by other regulatory components such as predicted peptidase cascades. A subset of each these classes of systems included several other linked genes beyond the core components, which were either additional effectors or other sensory or regulatory components.

We could categorize the final set of systems into three broad categories: (1) The MoxR-vWA-centric ternary systems; (2) GTPase-centric systems with or without the tubulin and HSP70 components; (3) systems with other regulatory components such as peptidase cascades (*Figure 1A*). Below we describe in detail the systems belonging to each of these categories along with the special 'grammatical' features of each of them.

## System 1: The MoxR-vWA-centric ternary systems

These systems are characterized by a core chaperone-co-chaperone pair of the AAA+ MoxR protein and a vWA domain protein. There are four distinct systems with these components: (1) the VMAP systems: these are characterized by presence of a third distinct conserved protein we termed the <u>v</u>WA-<u>M</u>oxR <u>a</u>ssociated <u>p</u>rotein (VMAP). These are by far the most frequent in the *nr* database. (2) The inactive STAND (iSTAND) NTPase systems: other than vWA-MoxR couple, these possess a catalytically inactive version of the STAND subfamily of AAA+ NTPases as the third component. (3) The FtsH-containing systems. These feature a further AAA+ ATPase of the classical AAA+ clade, FtsH, in addition to the core chaperone-co-chaperone pair. (4) The β-propeller-containing systems. These systems are characterized by a β-propeller domain fused to the vWA component. Beyond the ternary core a subset of each of these systems contain several additional effector and regulatory components.

We first discuss the two shared components of these systems: the MoxR AAA+ and the vWA protein and then discuss the unique features of each of the four systems separately.

## The MoxR and vWA components

The MoxR clade of AAA+ domains is defined by a distinctive insert in helix-2 of the core AAA+ domain (*Figure 1B*) and includes the dynein-midasin, MoxR, YifB and chelatase sub-clades. Across these sub-clades, with the apparent exception of dynein, the association with a vWA domain co-chaperone is observed (*Snider and Houry, 2006*; *Wong and Houry, 2012*; *Iyer et al., 2004a*). The vWA domain adopts an α/β Rossmannoid fold with six strands and has two conserved aspartates at the termini of the 1st and 4th core strands with which it typically binds a divalent metal ion like

$Mg^{2+}$. This $Mg^{2+}$-binding site of the vWA domain is part of the so called 'MIDAS motif', which mediates the binding of an extended or unstructured peptide (*Lee et al., 1995*; *Figure 1C*). Most of vWA domains found in the ternary systems we report here have the two aspartates and are likely to bind a peptide in a comparable manner. A subset of the vWA domains from the FtsH- and β-propeller- containing systems, however, appear to lack one or both aspartates (see *Figure 1—source data 1*). Hence, it is possible that they have acquired an alternative mechanism of peptide recognition.

The MoxR-vWA coupling has been extensively studied in several systems, such as: (1) the Midasin system involved in ribosomal protein-binding during pre-60S ribosome assembly in eukaryotes (*Chen et al., 2018*); (2) Archaeal phage-tail assembly system (*Scheele et al., 2011*); (3) the RavA-viaA system involved in the assembly of the fumarate reductase respiratory complex and the lysine decarboxylase complex in certain bacteria (*Wong et al., 2017*; *El Bakkouri et al., 2010*); (4) the YebA-coupled system likely involved in peptide-degradation (*Vollmer, 2012*); (5) the MoxR-MoxC/L and CoxD-CoxE systems involved in activating the insertase required for metal-cluster insertion in proteins (*Wong and Houry, 2012*; *Pelzmann et al., 2009*; *Fuhrmann et al., 2003*; *Pelzmann et al., 2014*); (6) the $Mg^{2+}$-chelatase involved in insertion of $Mg^{2+}$ into porphyrins (*Lundqvist et al., 2010*; *Fodje et al., 2001*). While these systems act in disparate biological contexts, a common biochemical denominator across all these is the capture of a peptide in the substrate by the vWA domain via its peptide-binding site, followed by ATPase action of the MoxR ATPase that transmits a conformational change via helical and flexible charged linker regions to the vWA component and the protein bound by it (*Snider and Houry, 2006*). This results in a conformational change in the latter with a consequent altered folding and release of the substrate: this process thereby contributes to macromolecular complex assembly (*Wong and Houry, 2012*).

In structural terms, known structures of vWA-MoxR pairs suggests that the MoxR AAA+ domain forms a hexameric ring typical of this superfamily of ATPases (*Wong and Houry, 2012*; *Maisel et al., 2012*). The MoxR ATPases found in the conflict systems under consideration possess the characteristic arginine finger N-terminal to the core strand-5 (*Figure 1B*), which is associated with ring-oligomerization (*Wong and Houry, 2012*); hence, they are likely to form a similar hexameric ring as other characterized MoxR proteins. While vWA co-chaperone domains can also form an oligomeric ring, their stoichiometry appears to differ with respect to the AAA+ units from system to system (*Chen et al., 2018*; *Scheele et al., 2011*; *Lundqvist et al., 2010*; *Liu et al., 2008*). Hence, it is unclear how many vWA units might associate with the MoxR ring in the systems we describe here. Based on the parallels to the known vWA-MoxR systems, we propose that even in these systems the vWA domains capture a peptide from a substrate protein and the ATP-dependent activity of the associated MoxR protein serves to release the bound substrate protein in an altered conformation. Given that these ternary systems are characterized by the further presence of at least one other protein in the complex, it is likely that this action of the vWA-MoxR pair helps restructure the complex formed with this additional component.

## vWA-MoxR subsystem 1: the distinguishing features of the core components of the VMAP-ternary systems

These systems display an unusual phyletic pattern being widely present in the actinobacteria, cyanobacteria, planctomycetes, and chloroflexi, and somewhat sparsely in alphaproteobacteria, gammaproteobacteria and deltaproteobacteria (*Figure 2*). Strikingly, there is a highly significant association between the presence of these systems and the presence of a multicellular or colonial or aggregate form (such as rosettes of planctomycetes and cooperating intracellular bacteroid aggregates with branching structures in rhizobia) in the life cycle of the organisms that possess them (*Figure 2*, *Figure 2—figure supplement 1*, *Table 1*). This is particularly notable in the lineages where they are sparsely found such as deltaproteobacteria, where they are only found in the multicellular myxobacteria or gammaproteobacteria, and where they are found in *Beggiatoa*, *Thiohalocapsa* and *Lamprocystis* which show multicellular or aggregative features (*Lyons and Kolter, 2015*; *Kysela et al., 2016*; *Figure 2—figure supplement 1*). The three core genes of this system show a strict preservation of gene order: from 5′ to 3′, the gene coding for the VMAP is followed by that for the MoxR, in turn followed by that for the vWA component. Comparison of the phylogenetic trees of these three components show a high degree of congruence across these systems (*Figure 3A*). We observed that 36% of the organisms with VMAP ternary systems code two or more distinct systems in their genomes each with their own set of three core components (*Figure 1—source data 1*). A record

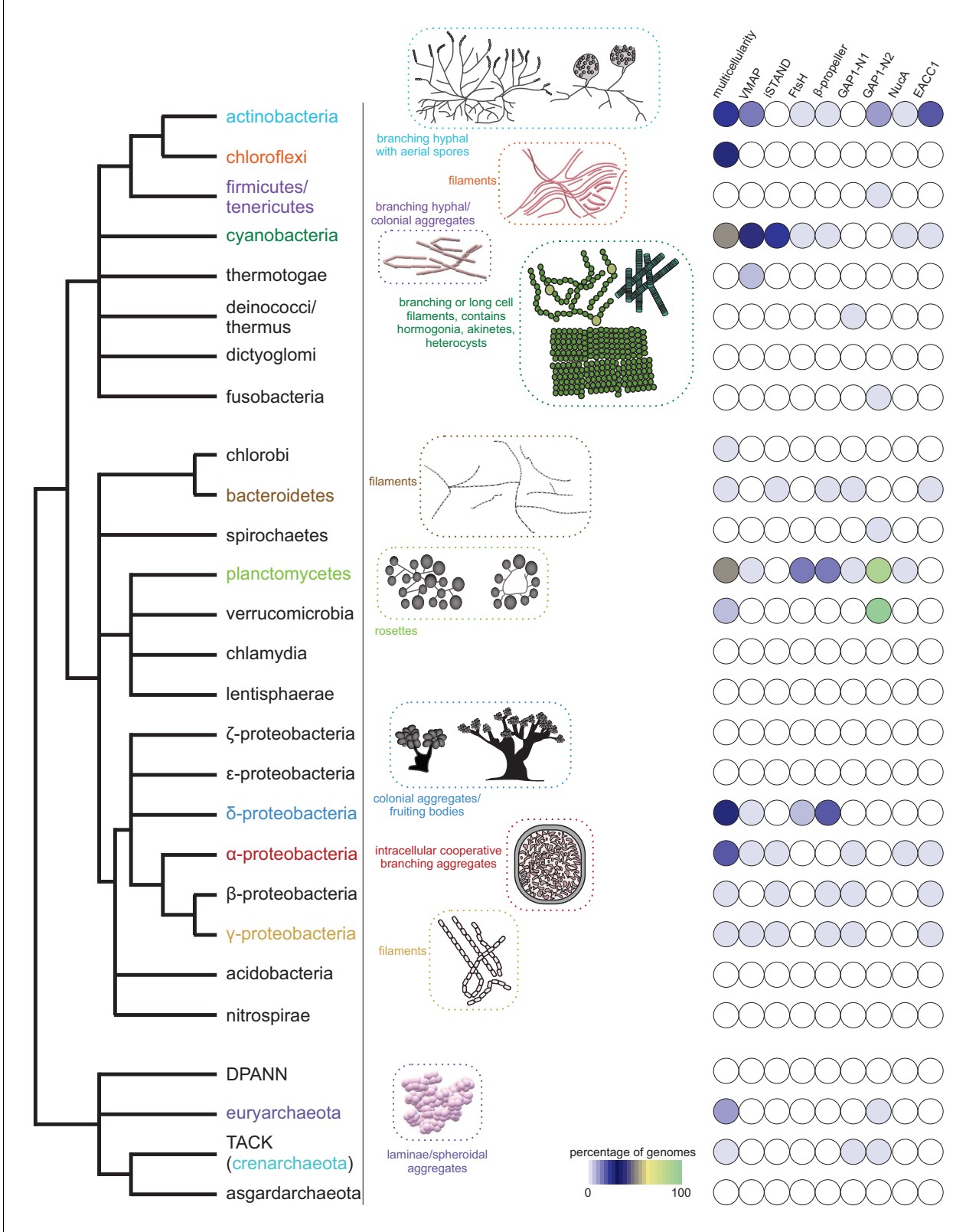

**Figure 2.** Emergence of multicellularity across prokaryotes and the presence of the newly-identified conflict systems. Tree of prokaryotic life (left) with general description and illustrations of known multicellular arrangements within clades (center) juxtaposed against presence/absence of the described systems (right). Coloring of clade labels matches multicellular descriptions. On the right, the fraction of organisms containing a system within a specific

*Figure 2 continued on next page*

*Figure 2 continued*

clade from the curated database (Materials and methods) is color-coded according to bottom legend. The first column on the right provides the fraction of known multicellular organisms within a clade in the curated database.

The online version of this article includes the following figure supplement(s) for figure 2:

**Figure supplement 1.** Hypergeometric distributions of the expected and observed number of multicellular prokaryotic organisms in each system type.

number of six paralogous systems are encoded by the actinobacterium *Streptomyces davawensis* JCM 4913. Together, these features suggest that the three core components of this system are under a strong selective constraint to function as a ternary complex with their own cognate partners and with a particular order of subunit assembly predicated by the conserved gene order. By analyzing the mean positional Shannon entropy across the alignment of the three core components, we observed that the vWA and VMAP components are evolving significantly more rapidly even between closely related species when compared to other conserved proteins in the respective genomes (e.g. replicative DNA polymerase β subunit) (*Figure 1E*). This is a characteristic feature that supports involvement of these systems in biological conflicts.

The MoxR-like proteins in these systems have a unique, poorly structured region N-terminal to the AAA+ domain found in no other members of the MoxR clade: this region displays two conserved motifs with multiple highly conserved aromatic residues and one nearly absolutely conserved arginine (respectively of the form: WxhapG and PsWRxa, where x is any amino acid, h is hydrophobic, a is aromatic, p is polar and s is small), and might adopt an extended conformation (*Figure 1B*). The vWA domain is flanked on both ends by well-conserved α-helical extensions with several strongly conserved residues. The N-terminal extension is typically separated from the vWA domain by a poorly structured linker region and shows a distinctive [D/E]xxx[D/E]xxxa motif. The C-terminal extension domain shows conserved S, E and R residues (*Figure 1C*). Beyond this C-terminal extension domain, the vWA components of these systems are marked by extraordinary variability, being fused to a succession of multiple further C-terminal globular domains. These can range in number from 1 to 11 distinct domains with an average of three domains, and they often greatly differ even between closely related organisms (*Figure 1F*). This feature again reinforces the involvement of these proteins in biological conflicts. Consistent with this, a detailed analysis of the domains in this region revealed parallels to those observed in other conflict systems (*Aravind et al., 2012*; *Burroughs and Aravind, 2016*; *Burroughs et al., 2015*; *Iyer et al., 2017*; *Zhang et al., 2012*; *Anantharaman et al., 2012*), suggesting that they are likely to act as effector domains with a diverse range of biochemical targets (*Figure 3A–B*; see below). The coupling of biochemically diverse effector domains is reminiscent of the pattern observed in the recently described polyvalent proteins deployed by phages and plasmids against their hosts (*Iyer et al., 2017*).

The third conserved component of these systems, the VMAP, is the most distinctive and it thus far not found outside of these systems. It has a tripartite modular organization (*Figure 3A–B*). Its most conserved part is a clearly distinguishable C-terminal domain (hereinafter VMAP-C), which is predicted to adopt a secondary structure with mixed α+β elements (a core of 7 β-strands and 6–7 α-helices) (*Figure 1D*) that could not be unified with any previously known domains. It features several, nearly universally-conserved polar residues (*Figure 1D*), suggesting a conserved interaction interface. While it cannot be entirely ruled out, there are no clear indications of an enzymatic function for VMAP-C. The central region of the VMAPs is highly variable and using sequence similarity-based clustering we were able to identify 29 distinct versions found in at least two or more distinct VMAPs (named VMAP-M0-28) (*Figures 3A–B* and *4*). Almost all of them are predicted to adopt an all-α-helical secondary structure, suggesting that they could all be rapidly diverging variants of a common α-helical fold (*Figure 1—source data 1*). They occur in widely different frequencies, with VMAP-M0 being the most common, occurring in 75% of the total proteins (1482) in our dataset, with the rest occurring much less frequently.

The N-terminal region of the VMAPs features one or more of 17 distinct domains belonging to three disparate classes (*Figures 3B* and *4B*):

1. A peptidase domain, most commonly either of the trypsin (*Rawlings and Barrett, 1994*) or of the caspase (*Aravind and Koonin, 2002*) superfamilies or on rare occasions of the transglutaminase-like thiol peptidase superfamily (*Anantharaman and Aravind, 2003b*). Whenever there

are trypsin-like peptidases, one of two genes frequently and mutually exclusively co-occur as neighbors. We named their protein products, which define hitherto unknown domain families, as trypsin-co-occurring domains 1 and 2 (Trypco1 and Trypco2) (*Figure 3B*). Whenever there is a domain of the caspase superfamily, there is additionally always a neighboring gene encoding a domain of the α/β-hydrolase superfamily, suggesting that the two operate together (*Figure 3B*). These peptidases might on occasions also be encoded by stand-alone genes adjacent to the core VMAP gene (*Figure 1—source data 1*).

2. An enzymatic domain generating a nucleotide-derived signal. The most common of these are members of the cyclic nucleotide (cNMP)-generating cyclase family (*Murzin, 1998*). On rare occasions we see the distantly-related cyclic di-nucleotide-generating GGDEF family domain (*Pei and Grishin, 2001*) instead of the cNMP cyclase. Alternatively, this region might feature members of the nucleoside phosphorylase superfamily or the TIR-deoxyribonucleotide hydrolase-SLOG fold, which we had predicted to be the generator of a second messenger molecule by releasing a base from a nucleoside/nucleotide in other conflict systems showing threshold-dependent regulation (*Figure 4A*, blue nodes) (*Burroughs et al., 2015*; *Samanovic et al., 2015*). This has indeed been demonstrated recently for the TIR domain, which generates ADP-ribose and its cyclic derivatives (e.g. cyclic ADP-Ribose (cADPR)) by cleaving nicotinamide from $NAD^+$ (*Burroughs et al., 2015*; *Burroughs et al., 2014*; *Anantharaman et al., 2013*).

3. A group of 10 distinct previously-uncharacterized non-enzymatic domains. When these are present at the N-termini of VMAPs a second copy of the same domain is usually found in the protein coded by the adjacent gene, where it is further fused to one of the above two groups of enzymatic domains or to an effector domain comparable to those found at C-termini of the vWA component (*Figures 3B* and *4B*; see below). We accordingly term these the Effector-associated domains (EADs, see below for discussion).

The enzymatic domains are strongly mutually exclusive of each other in VMAP architectures, but multiple EADs can occur in the same VMAP protein (*Figure 1—source data 1*). As a result of the different combinations of the above domains, we found a total of 71 distinct VMAP domain architectures from across 1482 distinct ternary systems of this type (*Figure 3B*).

## vWA-MoxR subsystem 1: the variable effector domains of the VMAP-ternary systems

We found a total of 163 distinct domain-architectures of the vWA component featuring 86 distinct C-terminal domains in our collection of 1482 VMAP ternary systems (*Figures 1F* and *3A–B*). A comparison of these ternary systems reveals an overarching theme: While about 15 effector domains are repeatedly observed, the rest tend to be rare and are found in less than 3% of the architectures. Comparisons between closely-related organisms suggests that this pattern emerges from effector domains continually displacing existing ones or accreting to existing architectures (*Figure 3A*). To better understand the action of these systems, we categorized the effector domains according to their biochemical functions. We briefly describe each class below (see also *Figure 1—source data 1*):

**Table 1.** p-Values for association of a given system with multicellular habit.
The total number of organisms with a given system and the number of organisms which score as multicellular are also provided.

| System | total # | multicellular # | P |
|---|---|---|---|
| VMAP | 716 | 694 | 0 |
| iSTAND | 103 | 77 | 3.895e-24 |
| FtsH | 79 | 63 | 7.08e-23 |
| beta propeller | 118 | 44 | 0.0094 |
| GAP1-N1 | 262 | 48 | *0.9998* |
| GAP1-N2 | 904 | 601 | 6.006e-152 |
| NucA | 45 | 29 | 6.331e-08 |
| EACC1 | 671 | 650 | 0 |

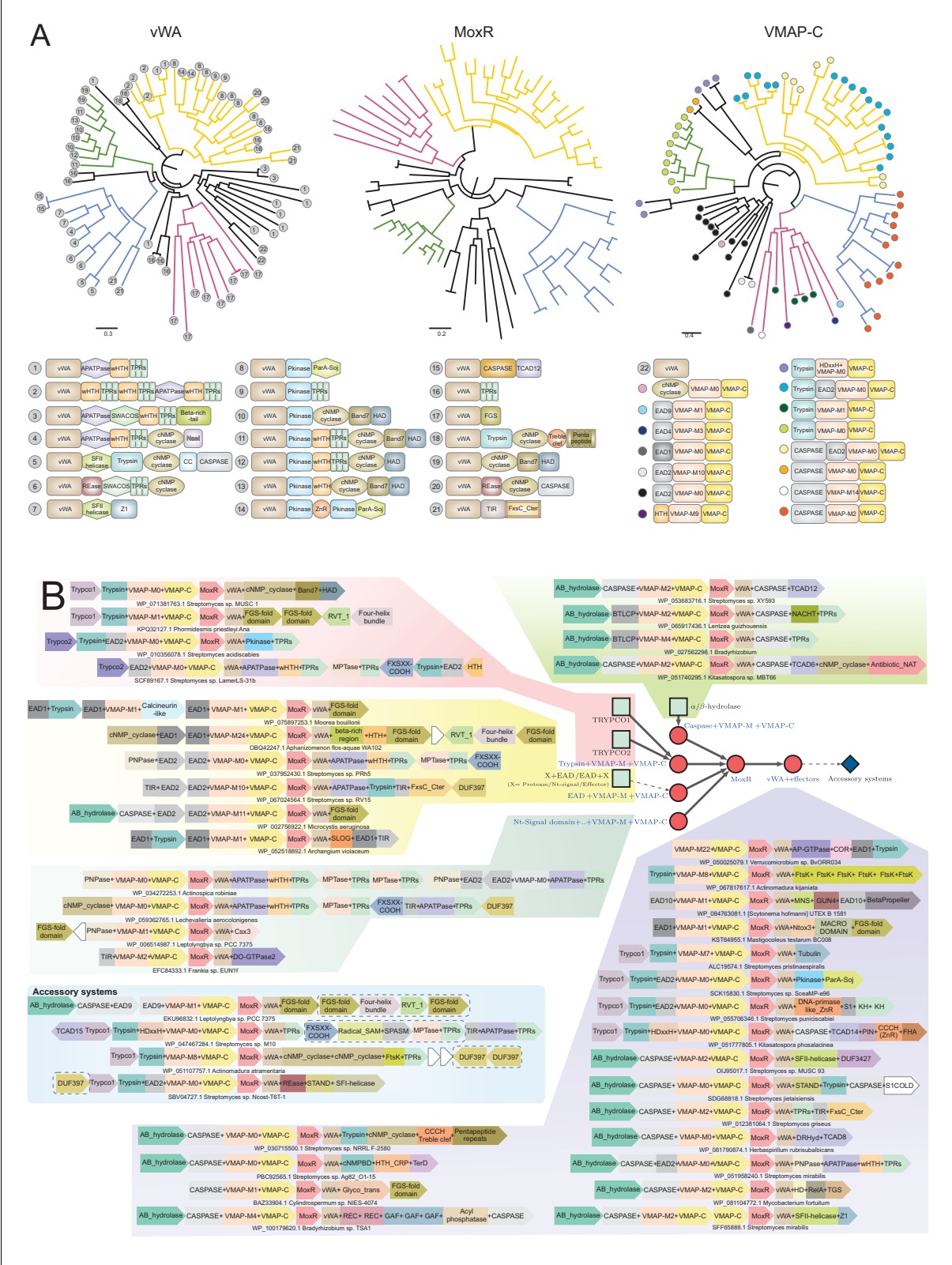

**Figure 3.** Diversity of classical ternary systems. (**A**) Evolution of the core components of the classical ternary systems across a common set of genomes. Scope of domain architectural diversity for the genomes under consideration are provided below for the vWA and VMAP-C components, linked to their respective trees by numbering (vWA) or color-coding (VMAP-C). (**B**) Generalized contextual diagram of the core components of the VMAP ternary system. The basic components are shown as nodes connected by arrows based on their gene neighborhood organization. The

*Figure 3 continued on next page*

*Figure 3 continued*

arrowhead points to the component encoded in the 3' direction on the genome. Core conserved components are shown as red circles. Example gene neighborhoods are provided, linked to distinct subtypes of the ternary system by background coloring. Poorly-conserved genes appearing in a neighborhood are unlabeled and depicted in white boxes. Accession numbers and organism names provided as labels below each neighborhood.

## Domains targeting nucleic acids

Across a range of biological conflict systems, the most common effectors are those that target nucleic acids, typically nucleases that either cleave nuclear DNA or RNAs involved in the translation process (*Aravind et al., 2012*; *Leplae et al., 2011*; *Burroughs and Aravind, 2016*; *Ishikawa et al., 2010*). Notably, in the systems under consideration, while such effectors are not very numerous (only found in ~7% of systems) there is a considerable diversity of them (18 distinct domains). The most common of these are enzymatic domains of the Restriction endonuclease (REase) fold, which specifically belong to the distinctive NaeI-like clade (*Steczkiewicz et al., 2012*), the Mrr-like clade or a previously unreported REase clade which we identified for the first time in these systems (*Figure 3B*). Notably, these systems also feature superfamily-II helicase partners of REases in classical R-M systems: (1) A member of the SWI2/SNF2-clade linked to a C-terminal Z1C DNA-binding domain and comparable to the H subunit of the CglI-like restriction enzymes (*Iyer et al., 2008*; *Toliusis et al., 2018*; *Figure 3B*); (2) a helicase related to those found in the EcoERI-like helicases type-I restriction systems. This is fused to a so called DUF3427 (Domain of Unknown Function 3427 [*Bateman et al., 2010*]) (e.g. *Streptomyces sp.* OIJ95017.1), which we and others predict bind modified DNA-based on the related domains found in the type-IV restriction enzyme, SauUS1 (*Xu et al., 2011*; *Weigele and Raleigh, 2016*; *Jablonska et al., 2017*; *Lutz et al., 2019*; *Figures 3B* and *4A*). Beyond these there are non-enzymatic potential DNA-binding domains in the effector module, such as different Helix-turn-helix (HTH) domains including those belonging to the CRP-like winged HTH family (*Aravind et al., 2005*; *Figure 3B*).

We also observed RNase domains that are likely to target RNA similar to those from toxin-antitoxin and polymorphic toxin systems. These include the PIN metal-dependent RNase domain (fused to a C-terminal Zn-binding domain) (*Figures 3B* and *4A*) and Ntox3, a metal-independent RNase domain from polymorphic toxin systems (*Zhang et al., 2012*; *Figure 3B*). We also identified a combination with the Csx3 domain from CRISPR/Cas systems (e.g. *Leptolyngbya sp.* PCC 7375; acc: WP_006514987.1) (*Figure 3B*). Csx3 is a RNA deadenylase in systems (*Yan et al., 2015*) that through structural comparisons we unify to the STAS domain fold (*Aravind and Koonin, 2000*) (LMI and LA unpublished observation), thereby establishing it as the third distinct RNase fold in these systems. As with the REases, we also found helicases among the effector domains that might act on RNA in the form of a superfamily-I helicase module, typically accompanying the PIN domain. Beyond these there are some RNA-binding domains, such as those with the OB-fold (ribosomal protein S1-like and the Cold Shock domains) (*Jiang et al., 1997*; *Keto-Timonen et al., 2016*), the KH (*Siomi et al., 1993*) and TGS domains (*Figure 3B*; *Wolf et al., 1999*). The S1-like and KH domain combination is related to the cognate domains found in the NusA protein (*Shin et al., 2003*), suggestive of a role in the context of transcription elongation.

## Peptidase domains

In addition to peptidase domains associated with the VMAP component, there are also peptidase domains which are fused in the 'effector-position' directly to the vWA component in about 9% of these systems. Here again, the trypsin- and caspase-like peptidases are the most frequently-observed domains and display a mutual exclusivity (*Figure 3B*). On rare occasions, there might be a metallopeptidase (MPTase) domain among the effector domains (*Figures 3B* and *4A*; *Stöcker et al., 1995*). However, unlike the VMAP-associated peptidase domains, ~70% of caspase domains and nearly all the trypsin domains in the effector position are catalytically inactive (*Figure 1—source data 1*). Hence, in stark contrast to the peptidases found in other biological conflict systems, where their action is likely predicated by their catalytic activity, in these systems they appear to function either as dominant-negative regulators or merely as peptide-binding domains.

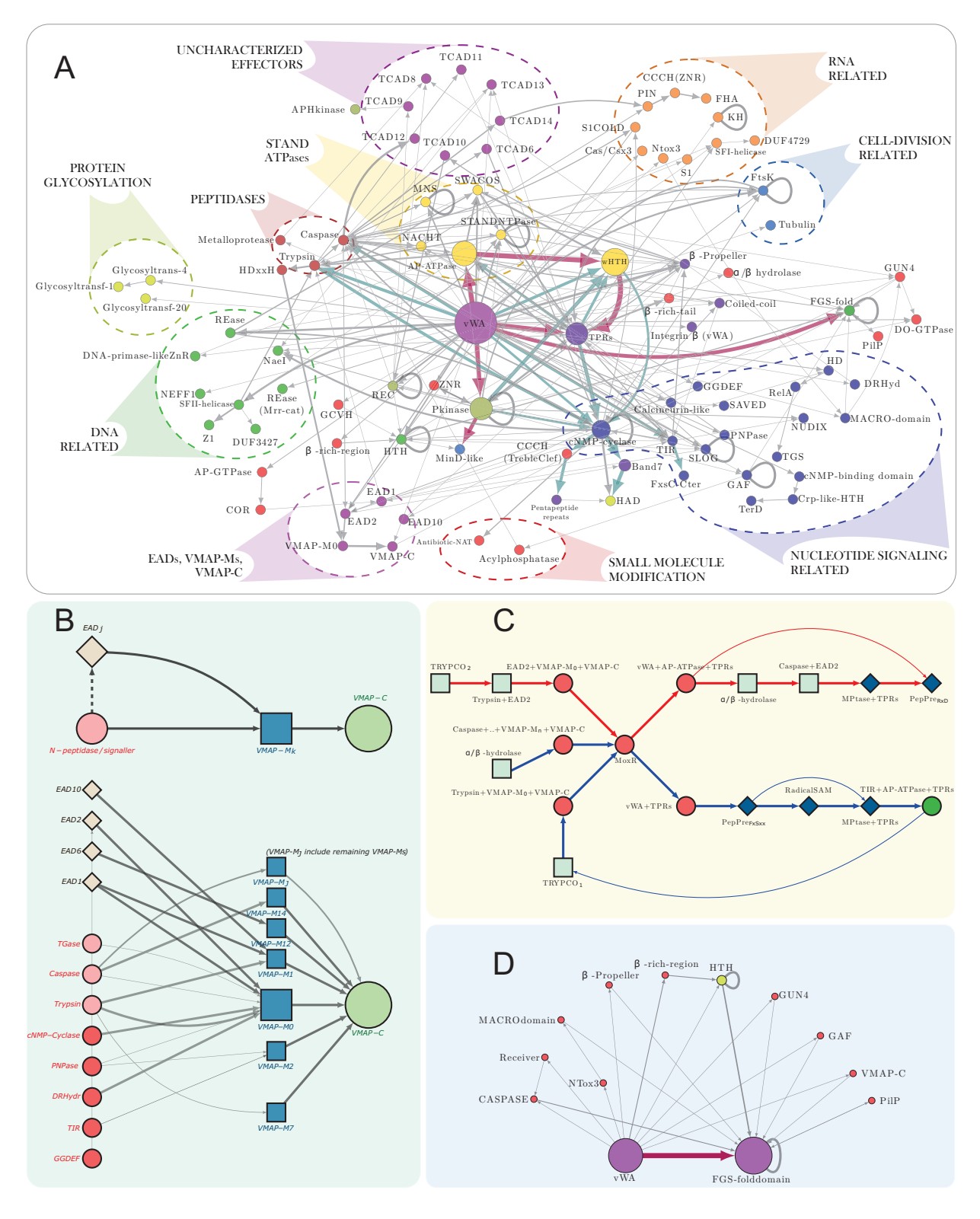

**Figure 4.** vWA architectural network and generalized classical ternary system diagrams. (**A**) vWA domain architecture network of the MoxR-vWA-centric ternary systems. Domains linked in the same polypeptide are connected by arrows with the arrowhead pointing to the C-terminal domain. Node size is scaled based on the relative frequency of occurrence of the domain in the systems and edge thickness is scaled based on the relative frequency of edge occurrence. Edges with >148 occurrences are colored maroon, whereas those with >14 connections are shown in cadet blue, others are shown in

*Figure 4 continued on next page*

*Figure 4 continued*

grey. Functionally similar domains are identically colored as follows: blue, nucleotide-dependent signaling; purple, adaptor, superstructure-forming, and structural; dark red, peptidase; dark-green, DNA-targeting; light blue, cell division-related; yellow, apoptosis-related; orange, RNA-targeting; light green, macromolecule modification; red, miscellaneous effector. Further these have been grouped together to the extent possible and indicated on the network. (B) Generalized contextual diagram of VMAP architecture, as described in *Figure 3B*. (C) Detailed contextual diagram of the MoxR-vWA-centric ternary systems, depicting mutual exclusivity of components of peptide-modification accessory systems and the Trypco-trypsin and α/β hydrolase-CASPASE peptidase pairings. See 2B for convention. Arrow colors reflect the distinctness of the contexts. (D) FGS domain architectural subnetwork of the VMAP ternary systems depicting its diverse domain associations.

## Nucleotide-dependent signaling domains

Approximately 20% of these systems feature one or more of 16 distinct domains that perform different roles related to nucleotide-dependent signaling. Whereas such domains are common in signaling systems (*Ponting et al., 1999*), only recently their roles in an effector capacity in biological conflict systems has come to light (*Burroughs et al., 2015*; *Iyer et al., 2006*). These can be further categorized into at least four broad categories: (1) nucleotide or nucleotide-derived second messenger-generating domains. The most common domains of this are the cNMP cyclases that are also associated with the VMAP component (*Figure 3B*). As with the peptidases, and in contrast to the cNMP cyclase domains found fused to the VMAP component, several of the versions fused to the vWA component are catalytically inactive. This again raises the possibility that they merely bind a nucleotide or act as a dominant-negative regulator. Less frequently, other nucleotide-generating domains like the GGDEF domain which generates cyclic di-NMPs (*Pei and Grishin, 2001*; *Ryjenkov et al., 2005*; *Schaap, 2013*) and the RelA nucleotidyltransferase domain, which generates ppGpp or the alarmone (*Magnusson et al., 2005*; *Mittenhuber, 2001*) are also observed (*Figures 3B* and *4A*). Another distinct enzymatic domain that is shared with the VMAP component is a member of the purine nucleoside phosphorylase (PNPase) superfamily which might generate a second messenger by releasing a base from a nucleotide/nucleoside (*Figure 3B*; *Burroughs et al., 2015*; *Burroughs et al., 2019*). (2) Nucleotide-binding domains. Overall, these are less frequently found than the nucleotide-generating enzymes and include multiple structurally unrelated domains such as the GAF, cNMP-binding domains (cNMPBD) and the more recently described SAVED and TerD domain, which might recognize oligonucleotides or specialized nucleotide derivatives, respectively (*Burroughs et al., 2015*; *Anantharaman et al., 2012*; *Aravind and Ponting, 1997*; *Ho et al., 2000*; *Figure 3B*). (3) Enzymes cleaving cyclic nucleotides, such as a calcineurin-superfamily phosphodiesterase (*Figure 3B*; *Aravind and Koonin, 1998b*). (4) Enzymes utilizing $NAD^+$ to generate derivatives like the ADP-Ribose (ADPr) and cADPr. Most prominently these include the TIR domains (*Essuman et al., 2018*; *Essuman et al., 2017*; *Figure 3B*) which were acquired on multiple independent occasions by the vWA components in actinobacteria, cyanobacteria and deltaproteobacteria from the versions fused to AP-ATPase domains (*Burch-Smith and Dinesh-Kumar, 2007*; *Leipe et al., 2004*). The DRHyd and SLOG domains which are evolutionarily related to the TIR domains (*Burroughs et al., 2015*) might also function in comparable capacity and are seen among the vWA-associated effector domains. Other enzymatic domains that might act on ADPr or directly on $NAD^+$ are the Nudix and Macro domains, with both being previously associated with $NAD^+$-linked biological conflict systems (*Aravind et al., 2015*; *de Souza and Aravind, 2012*; *Figures 3B* and *4A*).

## Domains related to macromolecule modification

About 18% of the systems feature effector domains related to modifications of proteins and other macromolecules. The most common of these are the serine/threonine/tyrosine (S/T/Y) protein kinases, which are found in the systems from actinobacteria and on rare occasions in chloroflexi (*Figure 3B*). Keeping with the vast expansion of such kinases previously noted in these multicellular bacteria (*Treuner-Lange, 2010*; *Aravind et al., 2003*), a phylogenetic analysis of the kinase domains in these systems show that they are specifically related to actinobacterial versions fused to either β-propeller domains or the cell-wall binding PASTA domains (*Yeats et al., 2002*) and have been acquired on at least two independent occasions by these systems. Remarkably, most of the kinase domains found in these systems show disruptions of the catalytic residues suggesting that they function primarily in a non-enzymatic role. However, we observed a few examples of a distantly related,

active kinase domain of the aminoglycoside-like (APH) kinase superfamily (*Hon et al., 1997*), which might phosphorylate low-molecular weight substrates. We also sporadically found other domains associated with phosphorylation-based signaling among the effectors such at the FHA domain which recognizes phosphorylated S/T containing peptides (*Durocher and Jackson, 2002*) and the receiver domain, which receives the phosphoryl group as part of the histidine kinase signaling system (*Pao and Saier, 1995*; *Figures 3B* and *4A*); however, we found no histidine kinase domains among the vWA-associated effectors.

At least three distinct families of glycosyltransferase domains (*Lairson et al., 2008*) and a GCN5-like N-acetyltransferase domain (*Neuwald and Altschul, 2016*) are also found in these systems (*Figures 3B* and *4A*) and might modify either protein substrates or other macromolecules such as peptidoglycan or teichoic acids (*Neuwald and Altschul, 2016*; *Caveney et al., 2018*). One of the versions of the inactive kinase domains and cNMP cyclase domains are often associated with a highly derived version of the Haloacid-dehalogenase (HAD) domain (*Burroughs et al., 2006*; *Figure 3B*). Given the lack of certain key active site residues such as a lysine, it is possible that these domains are catalytically inactive. Interestingly, they appear to have been derived from more widely occurring actinobacterial versions where they are coupled in an operon with an APH kinase.

## Eukaryotic apoptosis-related domains

A remarkable feature that sets these ternary systems apart from other known and predicted prokaryotic biological conflict systems is the frequent presence (~33% of the systems; *Figure 4A*) of several domains which were previously reported in eukaryotic apoptotic systems. It was indeed previously observed that these were particularly common in bacteria with multicellular forms or complex developmental phases (*Koonin and Aravind, 2002*). These are primarily members of the STAND clade of NTPases of the AAA+ superfamily (*Leipe et al., 2004*). While in bacteria they have been primarily studied as NTP-dependent transcriptional regulators (*He et al., 2008*; *Danot, 2015*), they likely hew functionally closer to their eukaryotic counterparts in these systems (see below). The most common STAND NTPases in these systems belong to the AP-ATPase family, with those of the SWACOS and NACHT clades occurring more sporadically (*Figure 3B*) and have been acquired on several independent occasions by the ternary systems (*Figure 1A*). Another eukaryotic apoptosis-related NTPase domain associated with the vWA component is the AP-GTPase which is distantly related to the Ras-Rab-Rho-Ran-like small GTPases (*Leipe et al., 2002*; *Aravind et al., 2001*; *Wauters et al., 2019*; *Fransson et al., 2003*; *Bosgraaf and Van Haastert, 2003*). They are much rarer than the STAND NTPases and are found in the systems from cyanobacteria and verrucomicrobia (e.g. *Calothrix* sp. PCC 6303, accession number: AFZ01290.1 and *Verrucomicrobium* sp. BvORR034, accession number: WP_050025079.1). They are strictly associated with the C-terminal COR domain involved in dimerization (*Gotthardt et al., 2008*; *Figure 3B*).

## Superstructure-forming and scaffolding domains

About 70% of vWA proteins in our dataset have domains in the effector position that can generally be categorized as either supersecondary structure-forming repeats and/or domains that form subcellular scaffolding structures through multimerization. The supersecondary forming domains include the TPRs and HEAT repeats, pentapeptide repeats, β-propeller repeats and coiled-coil regions, while the Band7-like domains are likely to form superstructures through multimerization (*Tavernarakis et al., 1999*; *Gehl and Sweetlove, 2014*; *Das et al., 1998*; *Thirup et al., 2013*; *Figures 3B* and *4A*). The prevalence of these domains in the VMAP ternary systems, especially in conjunction with the above apoptosis-related domains, suggests that formation of multimeric assemblies or super-structures are likely to be a key functional feature of these systems. Versions of pentapeptide repeats have been suggested to act as inhibitors of DNA-binding proteins (*Hegde et al., 2005*); thus, some of these complexes could affect nucleic acid protein interactions.

## Domains related to cell-division

About 7.5% of the effector domains are related to bacterial cell division. The most common of these is a member of the SIMIBI division of GTPases that is specifically part of the MinD/ParA-like family that regulate the formation of the central cell-division plane in bacteria (*de Boer et al., 1991*; *Hayashi et al., 2001*). Less frequently we observe homologs of the FtsK ATPase domains, which are

DNA-pumping proteins involved in chromosome partition, and rare examples of the FtsZ/Tubulin superfamily (*Yu et al., 1998*; *Aussel et al., 2002*; *Iyer et al., 2004b*; *Ma et al., 1996*; *Figures 3B and 4A*). Notably, as with several of the other enzymatic domains in this system, both the MinD/ParA-like and FtsK ATPase domains are predicted to be inactive.

## Miscellaneous effector domains

Other predicted effector domains outside of the above functional categories are sporadically found in these systems. One such is the FGS (Formylglycine-generating enzyme sulfatase) domain with a C-type lectin fold that is found almost exclusively in the cyanobacterial VMAP-ternary systems (*Figure 3B*). The FGS domain is related to the cognate domains found in the reverse transcriptase-dependent diversity-generating systems and are often linked to those (50,143–145) (see below for further discussion). Another domain found only in cyanobacterial systems is the tetrapyrrole-binding GUN4 domain (*Figure 3B*), which regulates the activities of the Magnesium chelatase or Ferrochelate enzymes involved in the biosynthesis of chlorophyll and heme, respectively (*Larkin et al., 2003*; *Wilde et al., 2004*). The GUN4 domain originated in the cyanobacteria, where it associates in the same polypeptide with diverse domains such as kinases, caspases, trypsin and TIR (*Davison et al., 2005*). These contexts suggest that GUN4 tetrapyrrole-binding likely allosterically regulates a wide range of enzymes depending on intracellular tetrapyrrole concentration, thereby acting as a sensor of the photosynthesis status of an organism.

## vWA-MoxR subsystem 1: architectural themes of vWA-component-associated effector domains of the VMAP ternary systems

We next investigated if the domain architectural patterns of the C-terminal domains associated with the vWA component might throw light on the functions of these systems. We observed that when more than one VMAP ternary system is encoded by an organism, in the majority of the cases the C-terminal domains coupled to the vWA are different in each of the paralogous systems (*Figure 1—source data 1*). This suggests that the systems are selected for the diversity of their potential interactions. To examine the combinations of different effector domains in greater detail, we constructed an architectural network using the vWA components from a set of 1482 distinct ternary systems (*Figure 4A*). It revealed that combinations of effector domains belonging to different functional categories are frequently observed. For instance, the RNase PIN domain might be combined in the same polypeptide with the phosphopeptide binding FHA domain or the NACHT NTPase domain while the NaeI REase domain comes with the cNMP cyclases (*Figure 4A*). In cyanobacteria, the FGS domain is combined with at least 10 other functionally diverse domains (*Figure 4D*); however, there is a strict architectural syntax with all these additional domains always occurring to the N-terminus of the FGS domain (*Figures 3B* and *4A,D*). Taken together, these observations suggest that disparate domains might be coupled together in the vWA component in order to mediate a multiplicity of distinct interactions at the same time as observed in the recently-described polyvalent toxin systems (*Iyer et al., 2017*).

Interestingly, a plot of the length distribution of the vWA component pointed to certain preferred lengths: it shows a multimodal distribution with peaks around 850, 1125, 1300 and 1450 residues (*Figure 1G*). The first peak is dominated by proteins from cyanobacteria while the remaining three are enriched in actinobacterial proteins (*Figure 1—figure supplement 1*). A closer examination revealed that this pattern arises from certain persistent and over-represented architectures. For example, we found the cNMP cyclase domain to be frequently coupled to N-terminal trypsin-like and C-terminal Band seven domains. The Band seven domains in turn tend to be strongly coupled to a C-terminal HAD domain. Similarly, the S/T/Y kinase domain tends to be coupled with either a C-terminal MinD/ParA domain or a wHTH domain (*Figure 4A*, *Figure 1—figure supplement 2*). Hence, it is conceivable that in these cases, despite the domains being functionally disparate, their action might be coordinated during effector deployment.

While multiple second-messenger-generating and peptidase domains are shared by the vWA component and the VMAP component, they do not necessarily co-occur in the same systems. For instance, whenever there is a trypsin or a cNMP cyclase domain associated with the vWA component, they are never or rarely found associated with the cognate VMAP component. However, when there is a caspase-like domain in the vWA component, they are nearly always accompanied by a

caspase-like domain fused to the VMAP component and its associated α/β-hydrolase domain. This suggests that, at least in part, the same domain has a different functional significance depending on the component in which it is found.

## vWA-MoxR subsystem 2: the iSTAND ternary systems

These form the second class of MoxR-vWA ternary systems, which are distinguished from the above-described VMAP ternary systems in having a distinct third component in lieu of the VMAP. Using sequence searches as well as profile-profile comparisons, we were able to show that the conserved core of this third component is an inactive version of the STAND NTPase domain (*Leipe et al., 2004*; *Saraste et al., 1990*): accordingly, we call this protein the iSTAND protein (*Figure 1—source data 1*). While the total number of these systems are far fewer than the VMAP systems, paralleling the latter systems they show a statistically significant presence in organisms with a multicellular state in their life cycle (*Figure 2*, *Figure 2—figure supplement 1*, *Table 1*). Notably, they are concentrated in cyanobacteria, proteobacteria and bacteroidetes while being almost entirely absent in actinobacteria (*Figure 1—source data 1*). Thus, they are the dominant ternary systems in multicellular forms among proteobacteria and bacteroidetes where the VMAP systems are rarer or absent.

In terms of their core components, the iSTAND system MoxR lacks the specific N- and C-terminal extensions seen in the VMAP systems (*Figure 1B*). The vWA components of these systems tend to have a 4-helical N-terminal domain that can be unified with the 4-helical N-terminal domain found in the CoxE-type vWA-proteins (*Pelzmann et al., 2014*). Another distinctive feature of these systems is the presence of transmembrane I segments at the N-terminus in about 28% of examples of this system (*Figure 5A*). Here too, the C-terminal region of the vWA component is highly variable and features one or more effector domains (*Figure 1—source data 1*). Paralleling the VMAP systems, when multiple distinct iSTAND ternary systems are encoded by the same organism, the vWA-associated effector domains tend to be distinct in each of them (*Figure 1—source data 1*). The number of effector domains found in these systems are fewer than in the VMAP systems and the most common one is the FGS domain, which is found in about 22% of the examples (*Figure 5A*). Beyond this, the superstructure forming domains, caspase-like peptidase domains and the TIR domain are common with the VMAP systems (*Figures 3B* and *4A*). Instead of the AP-GTPases, these systems feature a GTPase effector domain of the AIG subfamily of the GIMAP-Septin-like clade (*Leipe et al., 2002*), which are implicated in several immunity-related processes in eukaryotes (*Poirier et al., 1999*; *Reuber and Ausubel, 1996*; *Figure 5A*). Importantly, these systems differ from VMAP ternary systems in possessing several peptidoglycan-associating effector domains such the PEGA, OmpA, PASTA, and Sel1-repeat domains (*Yeats et al., 2002*; *Bouveret et al., 1999*) that are typically preceded by a TM segment (*Figure 5A*). Together with the N-terminal TM segments, these features imply that, unlike the VMAP systems, a subset of iSTAND systems are likely to function proximal to the membrane and in some cases interact with the cell-wall (*Figure 1—source data 1*). This might also relate to their phyletic patterns, that is the concentration in organisms with Gram-negative cell walls (*Figure 2*).

While the iSTAND domain of these systems could not be unified with the VMAP-C, the iSTAND component shows clear architectural parallels to the VMAP. Like in the VMAPs, either nucleotide-derived second messenger-generating domains (TIR or DRHyd) or peptidase (caspase-like and trypsin-like) or 4 of the EAD domains found in the VMAPs are also fused to the N-termini of the iSTAND domains (*Figure 5A*; see below). This points to a similar mechanism of action for these domains in conjunction with the other components of the system. We found a distinct α-helical domain at the N-termini of a subset of the iSTAND proteins. By means of profile-profile searches, we were able to unify this domain with domains such as Death, DED, CARD and Pyrin which display the Death-like fold and function in animal apoptotic systems (*Liu et al., 2003*; *Park et al., 2007*; *Aravind et al., 1999*; *Figure 5A*). This is the first time a domain with the Death-like fold has been found outside of animals; this bacterial Death-like domain (bDLD1/EAD3) might function similar to the Death-like domains in animal apoptosis (see below).

## vWA-MoxR subsystem 3: the FtsH ternary systems

The third distinct class of MoxR-vWA-centric ternary systems, like the previous two, show a statistically significant presence in organisms with a multicellular stage in their life cycle

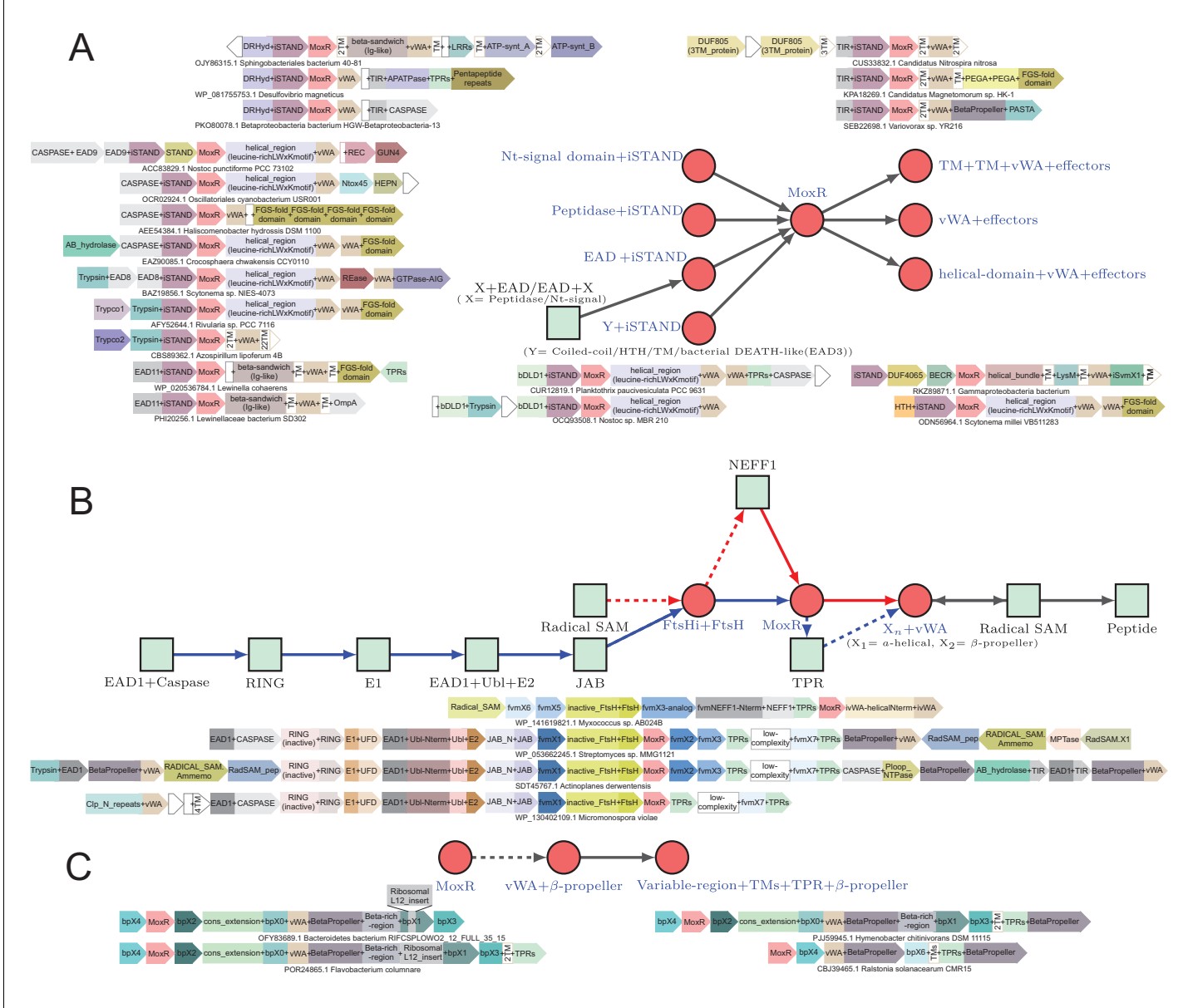

**Figure 5.** Generalized contextual diagrams and example genome contexts of other described ternary systems. (**A–C**) Generalized contextual diagrams for the components of the (**A**) iSTAND-, (**B**) FtsH-, and (**C**) β-propeller-containing systems. Coloring and connectivity as described in previous legends. Dotted lines reflect connections that are not universally present in a contextual theme. Example gene neighborhoods provided for systems as described in legend to *Figure 3B*.

from deltaproteobacteria, actinobacteria and planctomycetes (*Figure 2*, *Figure 2—figure supplement 1*, *Table 1*). Unlike the iSTAND and VMAP systems, they are rare in cyanobacteria (*Figure 1—source data 1*). This class is distinguished by the presence of a third conserved component with two copies of the FtsH-type AAA+ ATPase domain, with the N-terminal domain being catalytically inactive with loss of the P-loop and Walker B motif and the second domain possessing all the characteristics of active FtsH-type ATPases (*Okuno and Ogura, 2013*; *Figure 5B*). Further, a subset of these might have an additional N-terminal domain that is thus far seen only in these proteins. The vWA component of these systems is highly divergent and, in some cases, might even show a loss of the metal-chelating aspartates characteristic of the domain (*Figure 1C*, *Figure 1—source data 1*).

These systems can further be divided into two types based on the architecture of the vWA protein which occurs in the approximate proportion of 30% to 70% in the current *nr* database:

1. In the simpler of these systems, the vWA domain is fused to a N-terminal large all α-helical domain and the effectors are encoded by adjacent standalone genes. The only effectors seen in this subset of systems is a novel uncharacterized effector shared with the VMAP ternary systems, which we named NEFF1, and a distinctive enzymatic radical SAM domain (*Broderick et al., 2014*) which likely modifies a peptide-effector encoded by another standalone gene (*Figure 5B*).

2. The more complex variant of these systems usually shows an N-terminal β-propeller domain fused to the vWA domain, although rare variants are fused to repeats of the ClpN domain (typically found N-terminal of ClpAB AAA+ proteins [*Barnett et al., 2000*; *Figure 5B*]). A further distinguishing feature of the system is a complete prokaryotic Ubl-conjugation with separate genes coding for the JAB peptidase, E1, and prokaryotic RING-type E3 component and a protein that combines an Ubl with a E2 domain (*Burroughs et al., 2011*; *Burroughs, 2012*; *Burroughs et al., 2012*). Additionally, the Ubl is coupled to an N-terminal EAD1. Another gene in these operons encodes the effector, typically a caspase-like peptidase, which is also fused to EAD1 (*Figure 5B*). The presence of a tri-ligase Ubl-conjugation system along with JAB peptidase, which could release the Ubl from the multidomain architecture in which it is lodged (*Figure 5B*), suggests that the action of these systems is closely coupled to the conjugation of the Ubl to a certain substrate. These systems also additionally possess multiple α-helical proteins which are not found anywhere else outside of these. This, together with presence of the second AAA+ ATPase chaperone FtsH in these systems, suggests that these are likely to deploy their effector in the context of a large, multimeric complex with controlled (dis)assembly steps.

## vWA-MoxR subsystem 4: the β-propeller-containing ternary systems

The last group of ternary systems are found mostly only in bacteroidetes, planctomycetes, verrucomicrobia and proteobacteria (*Figure 1—source data 1*). While there is a significant tendency for these systems to be concentrated in forms with a multicellular habit, these are also found in certain bacteria that are not known to exhibit such an organization, for example plant-pathogenic *Pseudomonas* and *Xanthomonas* (*Figure 2*, *Figure 2—figure supplement 1*, *Table 1*). These systems are characterized by a vWA component that strictly possesses a β-propeller domain with about four blades C-terminal to the vWA domain. Further, about 62% of the vWA components are distinguished by a long N-terminal region with an uncharacterized α-helical domain and a C-terminal region with a predominantly β-strand-containing domain (*Figure 5C*). Remarkably, in several bacteroidetes, we find insertions of the multimerizing ribosomal protein L12/ClpS domain (*Mandava et al., 2012*) at different points C-terminal to the vWA domain (*Figure 5C*). The β-propeller-containing systems are enigmatic in that they show no conventional effector modules. However, their role in biological conflicts is underscored by the presence of a third conserved component with a highly variable N-terminal region (characteristic of conflict systems), which additionally typically has C-terminal TPR and β-propeller regions. This N-terminal variable region, which might function in the capacity of an effector, includes a globular domain and a membrane-spanning segment indicating that these operate close to the cell-membrane (*Figure 5C*). Given that there are β-propellers in both this third component and at the C-terminus of the vWA component, it is conceivable that it associates with the vWA component via homotypic interactions of these regions. A subset of these systems might feature additional, conserved linked genes coding for proteins that cannot be unified to known protein superfamilies. These proteins are predicted to contain unique domains, one of which is α-helical and others with mixed α+β elements (*Figure 1—source data 1*).

These systems share some features with the FtsH-containing systems despite lacking the FtsH component: First, both feature divergent vWA domains which might be coupled to β-propeller and other distantly-related domains which do not appear to be the effector module (*Figure 1—source data 1*). Second, the MoxR component of both systems often features a peculiar variant of the P-loop motif of the form GXXXTAKS (*Figure 1—source data 1*). This suggests that the two likely share a closer common ancestor, which diverged via acquisition of different effector modules and accessory components. Consistent with this, there is a 17% overlap in the organisms containing these two sets of systems (*Figure 1—source data 1*).

## Other systems linked to the MoxR-vWA-centric ternary systems

Operons coding for conflict systems often cluster together in the same genomic regions, potentially due to selection for shared regulation and concomitant deployment of these systems (*Burroughs et al., 2015*; *Anantharaman et al., 2012*; *Burroughs et al., 2014*). Sometimes this linkage is generic and does not persist between different species. However, in several cases, independent systems come together to form larger conserved neighborhoods that might act in a synergistic or an amplifying role (*Burroughs et al., 2015*). We found three distinct systems, which otherwise independently exist elsewhere, coupled to the ternary systems described above.

### Peptide-processing systems

About 20% of the actinobacterial VMAP-ternary systems and most of the FtsH-ternary systems (specifically those with the Ubl-tri-ligase components) are associated with a peptide-processing system. The core of the system is a gene coding for a zincin-like metallopeptidase (MPTase) and another encoding a small, rapidly-diverging protein (several of which are called FxSxx-COOH in Genbank). The latter protein is cleaved by the former peptidase to release a small C-terminal peptide (*Haft and Basu, 2011*). Although the precursor protein is highly variable, the C-terminal peptide is more conserved and based on the motifs in that peptide they can be divided into four types, namely: (1) the FxSxx motif type; (2) the RxD motif type; (3) the acidic type (containing four or more contiguous acidic residues); (4) the FtsH-associated type (*Figure 1—source data 1*). The first three are strictly associated with the VMAP-ternary systems. In the cases when the peptide has a FxSxx motif there is an additional gene coding for a radical-SAM domain further fused to a SPASM domain (*Benjdia et al., 2017*; *Figures 3B*, *4C* and *5B*). A distinct radical-SAM domain related to those found in the recently-described Memo-Ammecr1 systems is found in the peptide-processing systems associated with the FtsH-ternary system (*Burroughs et al., 2019*; *Figure 5B*). The radical-SAM domain catalyzes diverse reactions by reductively cleaving S-adenosyl-methionine to a highly reactive 5'deoxyadenosyl radical (*Broderick et al., 2014*; *Marsh et al., 2004*; *Mehta et al., 2015*). Streptide and sactipeptide synthesis provide the model for the action of the SPASM domain-fused radical-SAM enzymes in peptide-modifications – in those cases it respectively catalyzes intramolecular linkages between amino acids such as lysine and tryptophan or cysteine and other amino acids (*Bruender et al., 2016*). Accordingly, we predict a similar kind of modification of the peptides processed by the MPTase in the systems under consideration.

Whenever there is a peptide-processing system associated with the VMAP-ternary systems, there is an AP-ATPase effector (*Burch-Smith and Dinesh-Kumar, 2007*) with C-terminal TPRs (*Figures 3B* and *4C*). It is directly fused to the C-terminal region of the vWA protein if the peptide is of the RXD or acidic type. However, if it is of the FxSxx type, then the vWA component only has TPRs directly fused to it and the AP-ATPase with a N-terminal TIR domain is encoded by a separate gene occurring downstream of the peptide-processing system (*Figure 4C*). This suggests a close functional coupling between this particular type of VMAP-ternary system with AP-ATPase effectors and the peptide-processing system. This is supported by the existence of related standalone peptide-processing systems in actinobacteria, which also display a coupled gene coding for a similar AP-ATPase with an additional inactive MinD/ParA-like domain similar to those seen in some VMAP systems (see above, *Figure 3B*). Further, in some cases, these are linked to genes for a CRP-like cNMP-recognizing transcription factor suggesting that their action might be coordinated with sensing of a cNMP messenger (*Figure 3B*). Thus, it is likely that the processed peptide acts as a peptide toxin or signal that amplifies the action of the core system both in a subset of the VMAP- and FtsH-ternary systems (*Figures 3B*, *4C* and *5B*).

### Diversity-generating retroelements (DGR)

DGRs are a mechanism for generating diversity for anticipated ligand-binding in prokaryotes and phages that infect them (*Wu et al., 2018*; *Medhekar and Miller, 2007*; *Schillinger et al., 2012*). The core component of the DGR systems, prototyped by the system in *Bordetella* bacteriophage BPP-1, is a FGS domain protein, which displays enormous diversity in its C-terminal variable region (VR) (*Liu et al., 2002*). In the BPP-1 phage, this protein is part of the tail fiber that binds host receptors and undergoes anticipatory diversification allowing it to keep up with the evolutionary variation in the receptors due to selection for phage-evasion (*Doulatov et al., 2004*). The diversity is

generated by means of error-prone DNA synthesis by the reverse transcriptase (RT) component of the system, which interacts with the target nucleic acids aided by the third component of the system, a positively charged four α-helical bundle accessory protein (*Alayyoubi et al., 2013*). The RT of these systems is related to those found in group II introns, non-LTR elements and retrons (*Doulatov et al., 2004*; *Zimmerly and Wu, 2015*) and uses a template region (TR) that bears homology to the aforementioned VR for error-prone transcription such that several adenine bases are randomly mutated (*McMahon et al., 2005*). This DNA segment is then incorporated in place of the VR by the RT using adjacent constant sequences as guides causing non-synonymous substitutions in the protein (*Wu et al., 2018*). About 80% of the cyanobacterial and the gammaproteobacterial VMAP-ternary systems show an association with the diversity-generating retroelement (DGR) (*Figures 3A–B* and *4C*). In most of these cases, the vWA component is directly fused to the FGS effector domain, which shows an enormous sequence variability in its corresponding VR region (*McMahon et al., 2005*; *Le Coq and Ghosh, 2011*). These FGS domains are characterized by an all-β N-terminal extension that specifically relates them to the FGS domains of the hypervariable *Treponema denticola* TvpA DGR system (*Le Coq and Ghosh, 2011*). In the DGR-associated VMAP-ternary systems, we confirmed the presence of the TR segment, either present in the same gene neighborhood or elsewhere in the genome suggesting that indeed the observed diversity of the ternary system FGS is likely generated by the associated DGRs. This observation underscores the importance of diversification in the action of the vWA ternary systems.

## Linkage to components derived from DUF397 TA systems

The Pfam DUF397 is a small domain with four conserved β-strands and a terminal α-helix currently found only in actinobacteria. It is characterized by a strongly conserved CxE motif in the β-strand-2, and a RDS motif toward the end of β-strand-3. They are encoded in multiple copies by genes linked to about 4% of the VMAP-ternary systems. Outside of these systems, they are similarly found in multiple tandem copies, coupled in a mobile gene-neighborhood with a gene encoding an HTH protein, which is found elsewhere as the antitoxin component of TA systems (*Makarova et al., 2009*). This suggests that the DUF397 and HTH genes constitute a toxin-antitoxin (TA) system with the DUF397 acting as the toxin and the HTH the antitoxin (*Makarova et al., 2009*). When associated with the VMAP-ternary systems the copies of the DUF397 gene occur independent of the HTH-coding gene (*Figure 4B*). Thus, the domain appears to have been recruited from the TA systems as an additional co-effector for a subset of the VMAP-ternary systems. The small size of the DUF397 domain and the fact that it typically occurs as multiple copies both in the TA and VMAP-ternary systems suggests that this domain might act through the formation of multimeric complexes, perhaps involving cross-linking by conserved cysteines.

## System 2: the GTPase-centric systems

These systems are defined by mobile gene-neighborhoods that are widely distributed across bacteria and are also present in a few archaea; they are strikingly nearly completely absent in cyanobacteria. An organism might possess up to three paralogous versions of these systems in their genomes and 6% of the organisms with such systems have at least two of them (*Figure 1—source data 1*). These gene-neighborhoods are analogous to the MoxR-vWA systems in combining genes coding for two conserved regulatory components with highly variable components that bear the hallmarks of effectors (*Figure 6*). The two conserved components of these systems are: (1) GTPases of a previously unrecognized family of the TRAFAC clade (*Leipe et al., 2002*) with a conserved glutamate in their Walker B motif. In 38% of these systems they occur as two paralogous copies suggesting that they function as dimers. Accordingly, we named them <u>do</u>uble-GTPases (DO-GTPases). 60% of the DO-GTPases have 1–2 N-terminal zinc ribbons (ZnR) and about 13% of them have N-terminal TM segments. This, together with the presence of one or more TM-containing components in 67% of the systems, suggests that a major fraction of these systems is likely to function in proximity to the cell-membrane (*Figure 6*). (2) A previously undescribed protein, which we call the GTPase-associated protein 1 (GAP1). GAP1 is typically encoded by a gene downstream of a DO-GTPase gene and in some cases is directly fused to it. This protein is comprised of three clearly distinguishable globular domains, which we named GAP1-N, GAP1-M and GAP1-C as per their position in the protein (*Figure 6*). GAP1 occurs in two distinct subtypes that are readily distinguished by versions of the

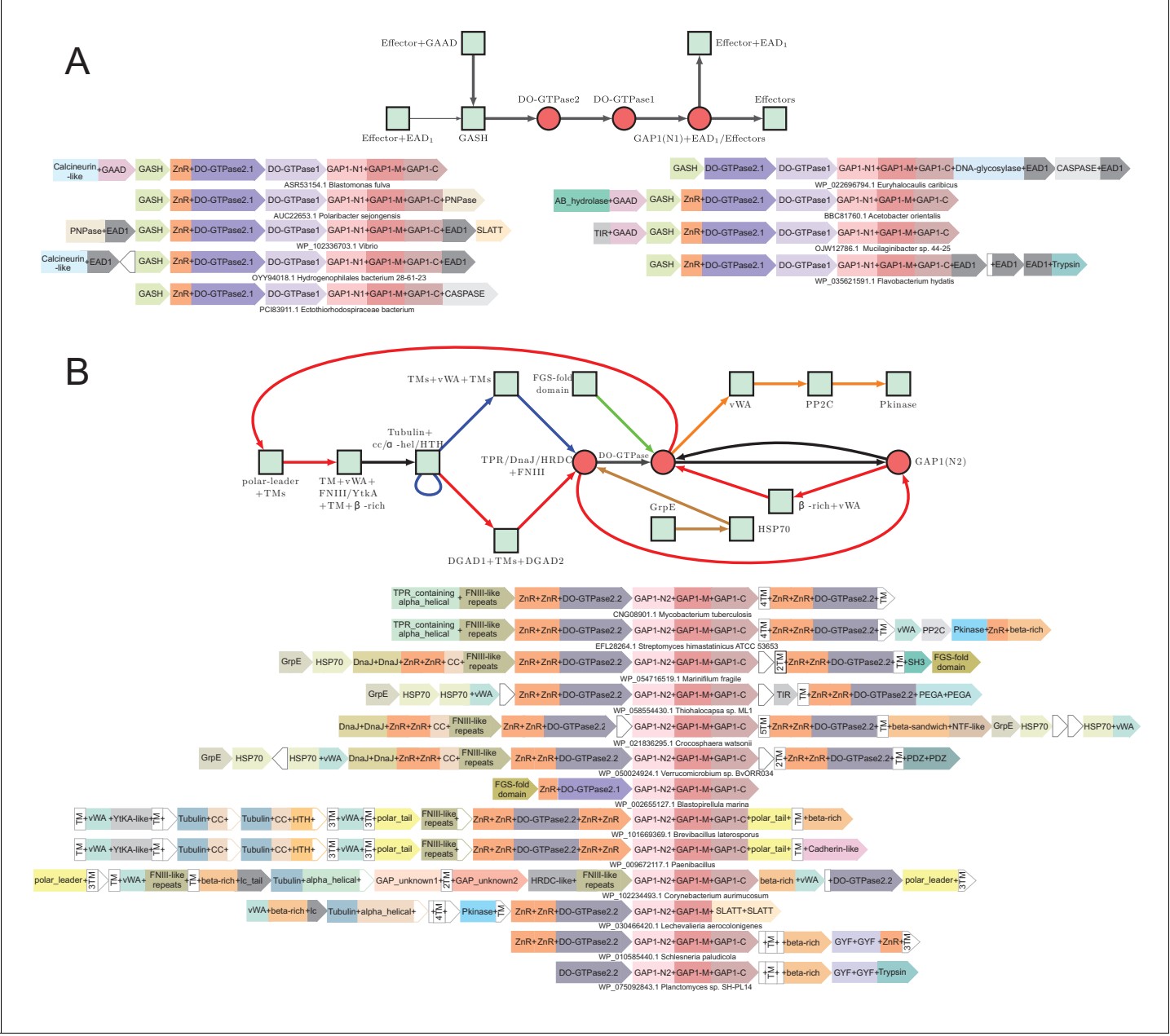

**Figure 6.** Generalized contextual diagrams and example genome contexts for the DO-GTPase systems. (**A–B**) Generalized contextual diagrams for the components of the DO-GTPase (**A**) GAP1-N1 and (**B**) GAP1-N2 systems. Coloring and connectivity as described in previous legends. Arrow colors reflect distinct contextual themes. Black arrows reflect connections present in more than one contextual theme. Example gene neighborhoods provided for systems as described in legend to *Figure 3B*.

GAP1-N domain (hereinafter GAP1-N1 and GAP1-N2) (*Figure 1—source data 1*). Notably, the associated DO-GTPases also show a sub-division into two basic types mirroring the type of GAP1 they occur with. These features indicate co-evolution between the DO-GTPase and GAP1 components and suggests that the GTPase dimer interacts directly with a cognate GAP1 component. Thus, we broadly classify these systems into two types based on the GAP1-N domain (*Figure 6A–B*).

Further, these systems contain either one of two types of mutually exclusive components with a distribution pattern that correlates with the version of the GAP1-N in the system (*Figure 6A–B*). Those systems with a GAP1-N1 domain (*Figure 6A*) contain a previously uncharacterized, all α-helical protein with a conserved domain which we name the GTPase-associated system helical (GASH)

domain. Those with a GAP1-N2 domain (*Figure 6B*) contain a protein with rapidly evolving repeats of the fibronectin type-III (FNIII) domain (*Little et al., 1994*). Beyond these, the systems contain one or more variable components: in some cases, these occur in the form of variable domains directly fused to the C-terminus of the GAP1 component. In other cases, they are fused to the N-terminus of the FNIII component. Additionally, these variable components might also be encoded by separate genes in the operon (*Figure 6B*). We discuss these below in the context of the two distinct subtypes of the GTPase-centric systems.

## GTPase-centric subsystem 1: *GAP1-N1-containing systems*

These systems are most commonly seen in proteobacteria, bacteroidetes and spirochaetes (*Figure 1—source data 1*). Unlike the vWA-centric systems described until now, these show no special association with a multicellular habit in the organisms possessing them (*Figure 2*, *Figure 2—figure supplement 1*, *Table 1*). In addition to genes for GAP1 and the GASH protein, these are characterized by two DO-GTPase genes, with one coding for a version with a N-terminally fused ZnR domain (*Figure 6A*). Notably, the variable effector domains in these systems almost completely overlap with those found in the VMAP- and iSTAND- MoxR-vWA ternary systems. The most common domain is the calcineurin-superfamily phosphoesterase domain predicted to act as nucleotidyl phosphodiesterase (*Aravind and Koonin, 1998b*). This is joined by other nucleotide-related effector domains such as the PNPase and the TIR domains (*Figure 6A*). On several occasions, when these nucleotide-related effector domains are present these systems also contain a gene for a 2TM SLATT domain protein (*Figure 6A*). This is notable because we had earlier predicted the SLATT domain to be a nucleotide-signal-responsive pore-forming effector in other conflict systems (*Burroughs et al., 2015*). Additional effectors in this system include the peptidases of the trypsin and caspase superfamily, and an α/β-hydrolase domain related to those found in the MoxR-vWA systems. On rare occasions, they may also feature a DNA-glycosylase domain (*Figure 6A*).

The variable effector domains appear to be coupled to the core components of these systems in at least three different ways. First, they can be directly fused to the GAP1 protein (*Figure 6A*). In other cases, the GAP1 protein contains a C-terminal EAD1 and the effector which is typically encoded by a further gene in the operon is also fused to an EAD1 suggesting that they might be brought together by a homotypic interaction of this domain (*Figure 6A*, see below). These architectures also suggest that the effectors of these systems are primarily deployed via an interaction with GAP1, probably mediated by the GAP1-C domain. The third and a frequent configuration combines the effector domain with a previously uncharacterized domain in the same polypeptide (*Figure 6A*). This small domain is predicted to adopt an α-helical fold and is not found outside of these systems (*Figure 1—source data 1*); hence, we predict that this domain is a likely adaptor domain that directly couples the effector to the GAP1 component. Accordingly, we named this domain the GTPase-associated adaptor domain (GAAD).

## GAP1-N2-type GTPase-centric systems

In contrast to GAP1-N1 gene-neighborhoods, the GAP1-N2-neighborhoods (*Figure 1—source data 1*), like most MoxR-vWA systems, show a highly significant association with organisms with a multicellular habit and are primarily found in actinobacteria, firmicutes, planctomycetes, verrucomicrobiae and chloroflexi (*Figure 2*, *Figure 2—figure supplement 1*, *Table 1*). These systems usually contain a GTPase component with two N-terminal ZnRs; if there are two GTPase components one additionally contains four N- and one C-terminal TM helices (*Figure 6B*). Additionally, some of these are fused to peptidoglycan-binding domains, like the bacterial SH3, PEGA or β-sandwich domains (*Ponting et al., 1999*; *Whisstock and Lesk, 1999*), or other ligand-binding (NTF2) and peptide-binding domains (PDZ [*Ranganathan and Ross, 1997*]) beyond the C-terminal TM segment (*Figure 6B*). Moreover, in these systems GAP1 itself can be frequently fused to C-terminal TMs (sometimes with further C-terminal cadherin domains) and other variable components also frequently containing TM segments (*Figure 6B*). These features indicate a role close to the cell-membrane for the GAP1-N2-type systems, with the extracellular domains suggesting potential interactions with the cell-wall or other extracellular ligands. Only a small number of effector domains of the GAP1-N2 systems are common to the GAP1-N1 systems and predominantly include the SLATT 2TM pore-forming effector domain, which is often fused to the C-terminus of GAP1 (*Figure 6A–B*). In planctomycetes,

we observed some systems with FGS domain effectors which are shared with multiple MoxR-vWA ternary systems, while others were coupled to a gene coding for a protein with a pair of GYF domains (*Kofler and Freund, 2006*; *Balaji and Aravind, 2007*) fused to either a trypsin or a TM-rich region (*Figure 6B*).

However, the majority of these systems have their own remarkable array of variable components that are mostly unlike those found in any of the systems discussed thus far. Some of these are fused to the FNIII-repeat component with others being encoded by arrays of linked genes in the neighborhood. These linked genes show lineage-specific patterns that can be divided into three non-overlapping themes. The first, comprising about 52% of all GAP1-N2 neighborhoods, is restricted to firmicutes and actinobacteria, and is centered on genes coding for proteins with tubulin domains (*Poirier et al., 1999*; *Reuber and Ausubel, 1996*; *Figure 6B*). The firmicutes versions are further distinguished by having two genes encoding proteins with tubulin domains fused to a C-terminal coiled-coil domain. These neighborhoods also contain two genes coding for proteins with vWA domains, one of which is fused to an uncharacterized domain called YtkA, while the other vWA-domain encoding protein has several TM helices with vWA in an extracellular position. Their GAP1 also has a highly-polar C-terminal extension (*Figure 6B*). The actinobacterial tubulin-containing systems display a single gene coding for a tubulin further fused to an uncharacterized C-terminal α-helical domain. These systems possess two distinct FNIII-repeat-encoding proteins, one fused to the DNA-binding HRDC-domain (*Morozov et al., 1997*) and the other is membrane-bound and fused to a vWA domain and an uncharacterized β-strand-rich domain (*Figure 6B*). Strikingly, the presence of these systems in actinobacteria is mutually exclusive of the VMAP-ternary systems suggesting a possible functional equivalence. The second theme, comprised of about 14% of GAP1-N2 neighborhoods is also predominantly observed in actinobacteria and firmicutes, and displays genes coding for HSP70 and its nucleotide exchange factor GrpE (*Bracher and Verghese, 2015*), with a subset of these containing a second gene encoding a protein with only the HSP70 domain sans its peptide-binding domain (*Figure 6B*). The FNIII-repeat-containing protein in these systems is typically fused to the HSP70 co-factor DnaJ domain (*Mayer and Gierasch, 2019*) or to TPR repeats (*Figure 6B*). The third theme, seen in 10% of GAP1-N2 neighborhoods is restricted to actinobacteria. This features an association with a catalytically active protein kinase domain often fused to a zinc ribbon and a β-strand-rich domain, or to TM helices. The gene coding for the protein kinase is usually linked to adjacent genes coding for a catalytically-active protein phosphatase domain of the PP2C family and a standalone vWA domain, respectively (*Figure 6B*). The systems point to regulated assembly of complexes akin to cytoskeletal structures and potential peptide-interactions via the vWA domains.

## The effector-associated domains (EADs) extend the repertoire of the above systems via homotypic interactions

We first uncovered 10 distinct EADs while analyzing the VMAP-ternary systems (see above), where they are typically fused to the VMAP component (*Figures 3B* and *4B*). Subsequently, we found a subset of them occurring in two of the remaining MoxR-vWA-centric ternary systems and the GAP1-N1-type GTPase-centric systems (*Figures 3A*, *4B*, *5A–B* and *6A*). The EADs are typified by the following distinct features:

1. They show a characteristic architectural pattern (*Figures 4B* and *7A*): One copy is always fused, typically to the N- or C-terminus, of a core component of the system under consideration, e.g. the VMAP, iSTAND, Ubl or GAP1 component of the respective systems. Further copies of the same EAD are fused to either effector or signaling domains, usually of enzymatic character, and are again either N- or C-terminally located. This second copy is often encoded by an adjacent gene in the same operon. Further copies might be encoded elsewhere in the genome but there is a strong correlation (where complete genomic sequences are available) with the presence of the cognate EAD fused to a component of the core system.
2. EADs are all small domains with no enzymatic features. Except for EAD4, EAD5, and EAD10, which have predicted mixed α+β character, all the remaining ones are entirely α-helical and several of them like EAD1, EAD6 and EAD7, are likely distantly-related as suggested by profile-profile searches (*Figure 1—source data 1*).
3. The copies of EADs which are encoded by adjacent genes tend to be more closely related to each other to the exclusion of others and more generally they tend to show closer relationships to lineage-specific versions.

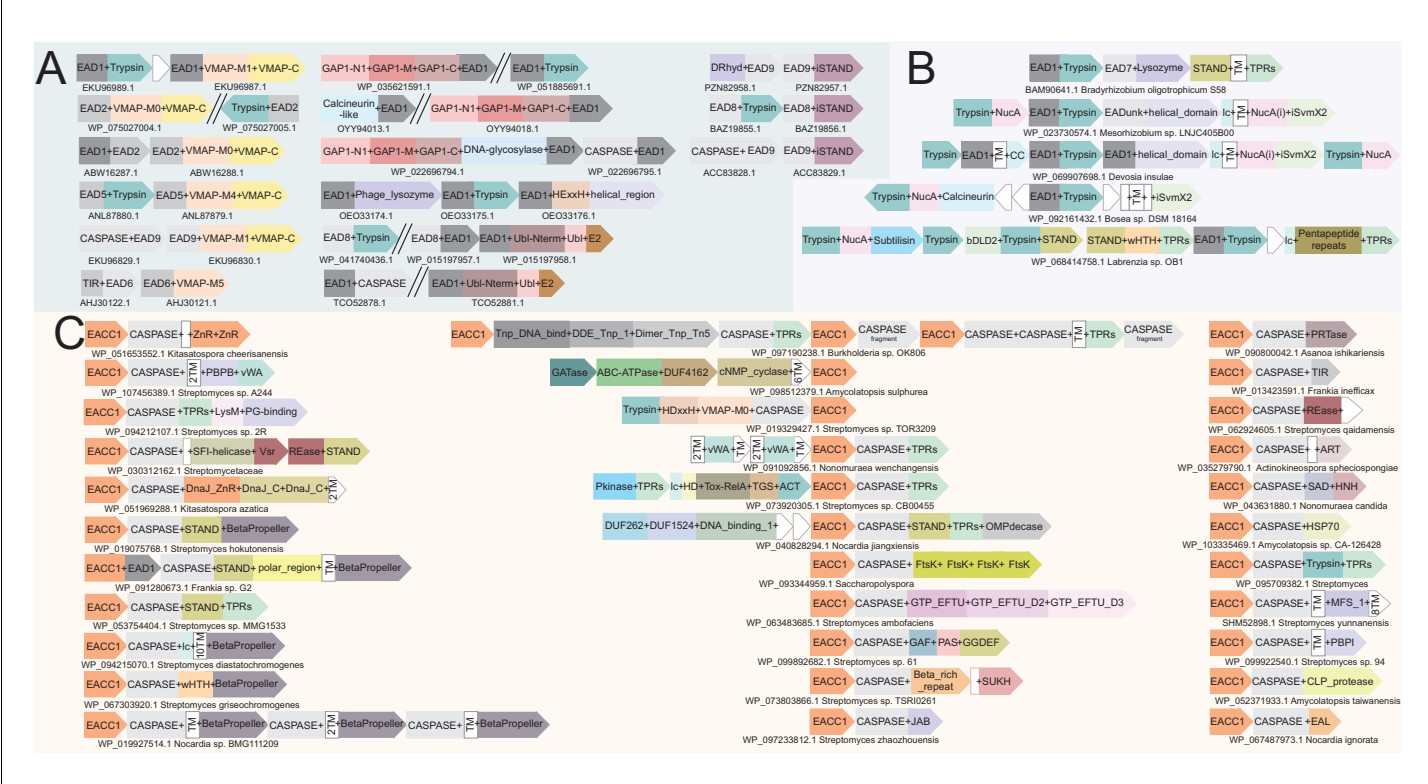

**Figure 7.** Example gene neighborhoods. (**A**) Illustrating the EAD-coupling principle, (**B**) of the NucA-peptidase system and (**C**) of the EACC1 two-gene system. Coloring and design as described in *Figure 3B*. EACC1 domain in (**C**) is colored in orange across systems, underscoring the conservation of the EACC1 component relative to the diversity of the associated CASPASE domain-fused component.

An organism typically encodes multiple proteins with the same EAD both linked to the primary systems and also elsewhere in the genome. The domains fused to them expand diversity of associated effector and signaling domains beyond those linked directly to the core systems. A record number of 22 distinct proteins with EAD1 are encoded in the genome of *Frankia inefficax* (*Figure 1—source data 1*), while the cyanobacteria *Mastigocoleus testarum* BC008 and *Scytonema hofmannii* PCC 7110 both contained 19 each. In the case of EAD2, the next most prevalent EAD, *Streptomyces davaonensis* JCM 4913 encodes eight paralogs. In cyanobacteria, EAD1-linked effectors add a considerable variety of effector and signaling domains to the ternary systems beyond the FGS domain (*Figure 7A*). The two most prevalent EADs, EAD1 and EAD2, show clear differences in phyletic patterns and architectural associations (*Figure 1—source data 1*): whereas EAD1 is found widely across bacteria with the above-described of systems, EAD2 is primarily found in actinobacteria and to a lesser extent in cyanobacteria. Whereas EAD1 is primarily linked to effector domains that also occur directly fused to the vWA component in MoxR-vWA ternary systems or GAP1 in the case of the GAP1-N1 GTPases systems, EAD2 occurs fused to the signaling peptidases and nucleotide-utilizing domains fused to the VMAP component of those ternary systems (*Figure 7A*). Thus, it appears that EADs like EAD1 (also bDLD, EAD4, EAD5, EAD7, EAD8, EAD10) primarily recruit other effectors to the systems, while those like EAD2 (also EAD6, EAD9) primarily recruit the signaling components of the system. Consistent with this distinction, there are multiple cases where the same VMAP component might contain both EAD1 and EAD2 fused to it (*Figure 7A*).

Notably, the duplicate presence of EADs in both the core component and the stand-alone components, the closer relationship to lineage-specific paralogs and the fact that the same effector or signaling domains might be directly fused to the core components in other cases, leads to the proposal that EADs are coupling adaptors. Thus, they are predicted to bridge effector and signaling domains to core components of different systems via homotypic interactions between themselves

(*Figure 7A*). A precedent for such a mechanism is presented by the animal apoptotic systems where α-helical domains (comparable to some of the EADs) of the Death-like superfamily (e.g. the Death domain, CARD, DED and Pyrin) have been implicated in homotypic interactions that result in polymeric effector complexes or oligomerization for the activation of the caspase effector domains fused directly to CARD domains (*Park et al., 2007*; *Kao et al., 2015*). Remarkably, as noted above, we found the first bacterial cognates of the Death-like superfamily (bDLD1/EAD3) occupying an equivalent position as the other EADs described here (*Figures 5A*, *7A* and *8A*). The implications of these parallels are elaborated further below in context of a mechanistic interpretation of these systems.

## Identification of other potential conflict systems that might deploy EADs

The observation that the EADs show a strong coupling, either in the same gene-neighborhood or in the same genome to diverse core systems with distantly or unrelated core components prompted us to investigate if they might lead us to other such systems. By systematically examining the domain-architectures and neighborhoods of the EAD-containing proteins we found three other potential conflict systems defined by characteristic gene-neighborhoods, which might occasionally or frequently utilize EAD1 or EAD2. We briefly describe these systems below.

### System 3: Coupling of multiple peptidase domains to NucA-like endonuclease domains

These systems are mostly found in alphaproteobacteria and are again significantly overrepresented in bacteria with a multicellular habit (*Figure 2*, *Figure 2—figure supplement 1*, *Table 1*). They are characterized by two core components, namely a NucA (Endonuclease NS)-type endonuclease domain of the HNH superfamily (*Zhang et al., 2012*; *Baslé et al., 2018*) and a trypsin-like peptidase, in several cases fused to TM segments. Both of these might be found fused together in the same protein or encoded in multiple copies in the same operon (*Figure 7B*). In some cases, a trypsin domain and a further peptidase domain of the subtilisin (S8) family (*Rawlings and Barrett, 1994*) might flank the NucA endonuclease domains in the same polypeptide. In a subset of these operons, one of the trypsin paralogs might be fused to an EAD1 domain. In these cases, there are other EAD1-containing proteins either in the same operon or in the same genome and are fused to a variable group of domains that potentially act as effectors. The EAD1-associated domains in the system include a lysozyme- and a papain-like peptidoglycan peptidase domain, pointing to a potential attack on peptidoglycan by these effectors. Certain operons further encode a STAND NTPase component, which is fused to an uncharacterized domain and a trypsin domain at the N-terminus (*Figure 7B*). Analysis of the uncharacterized domain using profile-profile searches unified it with the Death-like superfamily found in animal apoptotic proteins (*Park et al., 2007*; *Wilson et al., 2009*; *Martinon et al., 2001*), thereby making it the second family of Death-like domains to be identified in bacteria (bDLD2) (*Figures 7B* and *8A–B*).

Given that the organisms possessing versions of these systems with EAD1 encode no other system in the genome with EAD1, we propose that these represent a novel type of core system that recruits additional effectors via EAD1. This is supported by the fact that the core components are also observed in related organisms independently of EAD1. The trypsin-like peptidases and NucA are likely to comprise the essential core of the system that is probably activated via a proteolytic cascade to deploy the NucA endonuclease. The other components like the STAND NTPase with the Death-like domain or those brought in via the interactions of EAD1 are likely coupled effectors.

### System 4: a membrane-associated two-gene conflict system

This system is prevalent in actinobacteria and sporadically in proteobacteria with a phyletic distribution that again significantly comports with organisms having a multicellular habit (*Figure 2*, *Figure 2—figure supplement 1*, *Table 1*). The two genes comprising this system are (*Figure 7C*):

1. A 5′ gene encoding a domain we term the Effector Associated Constant Component 1 (EACC1). EACC1 is either a standalone domain or on some occasions fused to EAD1 or EAD2. Sequence analysis indicates that the EACC1 domain is comprised of a single conserved TM helix with short, N- and C-terminal extensions with α+β structural elements (*Figure 1—source*

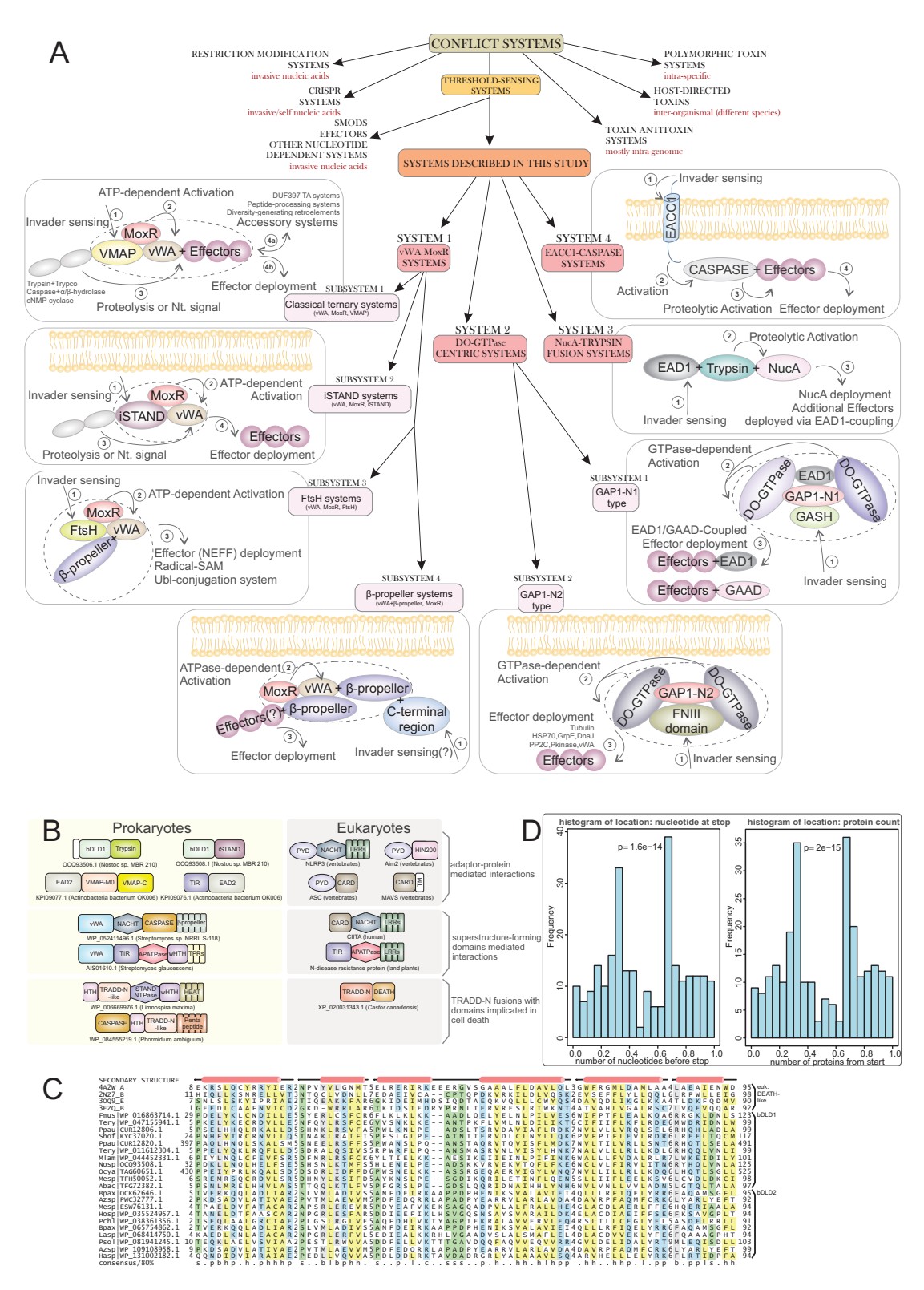

**Figure 8.** Systems categorization and connections to eukaryotic apoptotic systems. (**A**) Schematic representation and categorization of the systems described here placed in the context of other categories of biological conflict system classes. The various components of the systems described in this manuscript are depicted in their respective boxes along with a plausible mechanism of action, involving invader sensing, activation, proteolysis and effector deployment. The ternary core of each sub-system is delineated. (**B**) Cross-superkingdom comparison of the shared 'grammar' and domain

*Figure 8 continued on next page*

*Figure 8 continued*

content across the systems described herein (left column) and those experimentally-characterized in eukaryotic apoptotic pathways (right column). (C) Multiple sequence alignment of newly-identified bacterial Death-like domain families. Top four sequences are of known animal Death superfamily structures. (D) Histograms situating the VMAP ternary systems in actinobacterial genomes according to location of vWA domain-containing gene by stop nucleotide position relative to entire normalized nucleotide length of the genome (left) and protein ordering relative to normalized total protein count in the genome (right). Displayed p-values result from comparison to distribution expected by random genome distribution, using the $\chi^2$ test.

*data 1*). This suggests that it inserts into the membrane and protrudes beyond both the outer and inner layers of the bilayer.

2. A 3' gene that codes for a protein frequently possessing a caspase-superfamily peptidase at its N-terminus. In a striking parallel to the VMAP-ternary systems, a less-frequent architecture of this component involves displacement of the caspase domain by a cNMP cyclase domain (*Figure 7C*). At least 33% of all proteins carrying the caspase domain contain TM helices, suggesting that as a whole this system functions in proximity to the membrane.

Strikingly, we identified up to 50 distinct alternative domains belonging to a wide range of functional categories fused to the C-terminus of the constant caspase domain of the second component: these include many shared with the above systems, such as the peptidases, nucleotide-utilizing domains, signaling enzymes, STAND NTPases, superstructure-forming repeats, and a MoxR-independent version of the vWA domain (*Figure 7C*). However, we also observed several domains not seen in any of the above systems such as JAB- and Clp-like proteases (*Burroughs et al., 2011*; *Kress et al., 2009*), an OMP-decarboxylase-like, SAD modified DNA-binding (*Iyer et al., 2011b*), and Elongation Factor Tu domains (*Figure 7C*). There are also instances of TM-helices with the downstream extracellular periplasmic-binding protein (PBP) domain (*Mowbray and Cole, 1992*) in this position (*Figure 7C*). The large diversity of domains in this variable region is reminiscent of the C-terminal variable domains of the vWA component in several of the ternary systems (see above, *Figure 3A–B*) and points to a parallel deployment of these domains as effectors. The TM segment of EACC1 is strongly conserved and contains an unusual patch of small (often glycine) residues near the center of the helix (*Figure 1—source data 1*). This unusual attribute suggests that rather than being a mere membrane-anchoring segment, it might play a role as a membrane-associated sensor that then activates the constant caspase domain of the second component to in turn facilitate effector deployment via auto-proteolysis.

## The bacterial TRADD-N systems

The third class of systems are predominantly found in cyanobacteria and were identified by virtue of the linkages of EAD1 to a previously unrecognized domain. In related systems this domain also occurs independently of EAD1 as a constant module linked to a highly variable set of domains such as pentapeptide repeats, a caspase-superfamily peptidase, STAND NTPase and FNIII domains. Thus, these systems exhibit a similar organizational pattern as the above systems with a constant module coupled with a variety of modules that are rapidly varying between closely-related organisms. Through transitive sequence profile and profile-profile searches, we were able to unify the uncharacterized constant domain of these proteins with the TRADD-N domain. The core of the TRADD-N superfamily has an RNA-recognition motif fold with a highly conserved serine in the turn between $\beta-2$ and $\beta-3$ (*Tsao et al., 2000*; *Figure 8B*). In the animal apoptotic system, TRADD-N serves as an adaptor domain downstream of the tumor necrosis factor receptor and mediates protein-protein interactions (*Tsao et al., 2000*; *Micheau and Tschopp, 2003*). Thus, like many proteins of the apoptotic machinery, TRADD-N is likely to have had a provenance in bacterial systems with a role in conflict. As these systems have links to other hypervariable systems in eukaryotes, they will be discussed separately elsewhere (unpublished GK, AMB, LA).

## Discussion of shared organizational features pointing to unifying mechanistic principles related to biological conflicts

Although at first sight the above systems might appear rather disparate, they are unified by certain distinctive features that help reconstruct their potential mode of action:

1. They display a distinctive combination of core, constant modules (e.g. MoxR-vWA, DO-GTPases-GAP1) coupled with components that are highly-variable even between closely-related organisms, that is the effector modules. The general pattern of variability is reminiscent of other conflict systems and reflects a co-evolutionary arms-race scenario (*Burroughs and Aravind, 2016*; *Burroughs et al., 2015*; *Iyer et al., 2017*; *Krishnan et al., 2018*). However, the presence of the constant components is a distinctive feature that is not common to all conflict systems. Such a pattern is seen in the polymorphic toxin and eukaryotic CR-effector systems (*Zhang et al., 2012*; *Zhang et al., 2016*); in those cases, the constant components are specifically linked to the trafficking of the effector components through various secretory pathways. In contrast, in most of the systems under consideration here, there are no features that indicate secretion of primary effectors outside the cell, though they might often function on the inner side of the membrane. Hence, considering these factors, we may infer that these systems are primarily responding to entities that invade cells such as viruses and selfish elements like plasmids.

2. The constant components of most of these systems involve either an ATPase/GTPase element and/or a proteolytic element in the form of peptidase domains (most commonly trypsin- or caspase- superfamily peptidases). This suggests that the systems are likely in a constitutively 'off' state and need to be switched 'on' in response to the invasive entity. This also implies tight regulation of the effector and indicates potentially deleterious consequences to the cell if they were to be deployed without a stimulus. This is again typical of regulated conflict systems (*Aravind et al., 2012*; *Leplae et al., 2011*; *Burroughs et al., 2015*). Accordingly, based on known MoxR-vWA pairs (*Chen et al., 2018*; *Scheele et al., 2011*; *Wong et al., 2017*) we suggest that the effector-deployment step in the ternary systems involves MoxR ATPase-dependent reorganization of linker regions coupled to binding of specific peptides by the vWA component to allow re-organization of the complex in a new conformation. An analogous action is likely one part of the DO-GTPase systems; however, they are likely to act as a switch as with other GTPases, which transmits a conformational change on binding GTP and terminates the signal upon GTP hydrolysis (*Wauters et al., 2019*; *Bosgraaf and Van Haastert, 2003*). In either case, the result is a new conformation of the complex in which the effectors are poised to be deployed. At least in a subset of cases their final deployment is likely to require an additional event in the form of proteolytic processing by associated peptidase domains or via binding of a nucleotide-derived signal generated by the associated nucleotide-utilizing enzymatic domains (*Figure 8A*). Specifically, in the VMAP-ternary system, some examples of the DO-GTPase systems, the EACC1, and NucA-trypsin system the proteolytic action is likely to release the effectors to allow them to diffuse away (*Figures 3B*, *6A*, *7B–C* and *Figure 8A*). This might explain the tendency to couple multiple disparate effectors in the VMAP ternary system (*Figure 3B*). Such systems with multiple peptidase domains have analogs in the eukaryotic systems that activate a terminal signal or effector, such as the peptidase cascade in *Drosophila* dorso-ventral polarity establishment with multiple trypsin-like peptidases (*Dissing et al., 2001*) or the caspase-cascades in animal apoptosis (*Cohen, 1997*).

3. Deployment of such systems in response to invasive entities requires both the sensing of the invasive entity and the coordination of this sensing even with the rest of the apparatus. This appears to be related to the characteristic 'ternary' organization of most of these systems with a unique 'third' component such as the VMAP, iSTAND or GAP1 (*Figures 3B*, *5A* and *6A–B*). Being a constant component, they are likely to coordinate the sensing with switch components (e.g. MoxR, DO-GTPase). Notably, they are often fused to or tightly-associated with a highly-variable module, which, however, does not display any enzymatic features (e.g. the VMAP-Ms or the FNIII domains in the DO-GTPase systems) (*Figure 1—source data 1*). This implies that they are likely to be involved in directly interacting with the invasive entity to sense it. Sensing likely triggers a sequence of conformational changes, which are communicated to the switch-components via the more constant elements of the protein (e.g. VMAP-C, GAP1-N1/2).

4. The most remarkable element of these systems are the effectors. While they are variable as in other conflict systems (*Van Melderen, 2010*; *Burroughs et al., 2015*; *Iyer et al., 2017*; *Zhang et al., 2012*), they differ qualitatively: enzymatic domains that target nucleic acids or proteins shared with other conflicts systems are the minority in these systems. A unique feature of several of these systems is the presence of effector domains which are inactive versions of various enzymes that have active counterparts in the same cell (e.g. the VMAP-ternary systems) (*Figure 1—source data 1*). Such inactive enzymes have been previously encountered in eukaryotic immune systems (e.g. APOBEC3 deaminases [*Krishnan et al., 2018*; *Stavrou and Ross, 2015*; *Refsland and Harris, 2013*; *Aydin et al., 2014*]), which act as decoys that bind

viral proteins that would otherwise interact with the active copies. Also present are the hyper-variable FGS domains coupled to the DGRs (*McMahon et al., 2005*; *Le Coq and Ghosh, 2011*), which are not commonly observed as effectors in other common cellular conflict systems. These also appear to show the hallmarks of a direct interactor with molecules of the invasive entity without enzymatic targeting. Further, in the DO-GTPase systems, effectors are likely to form multimeric assemblies like the tubulins and HSP70 (*Ma et al., 1996*; *Desai and Mitchison, 1997*; *Wickner, 2016*; *Nogales et al., 1998*). The effectors are often coupled with EADs, which also show features implying homotypic interactions to form complexes by multimerization. Notably, among the most common type of effectors in these systems are either superstructure forming repeats or STAND NTPases and AP-GTPases that are encountered in both animal and plant immune systems responding to intracellular pathogens, animal apoptotic systems, and fungal hetero-incompatibility systems (*Leipe et al., 2002*; *Zhang et al., 2016*; *Leipe et al., 2004*; *Koonin and Aravind, 2002*; *Aravind et al., 2001*; *Glass and Dementhon, 2006*; *Danot et al., 2009*; *Seuring et al., 2012*; *Hofmann, 2019*; *Figure 8B*).

Thus, we may infer that the formation of multimeric assemblies is the common denominator of several of these systems: this has several parallels in eukaryotic apoptotic and anti-invader systems in terms of both shared components and regulation (*Figure 8A–B*). These eukaryotic systems are tightly regulated and kept in an 'off' state prior to being triggered by the direct sensing of the invasive entity or apoptotic signal. This usually happens by the direct interaction of the signal (usually a protein) with the C-terminal superstructure-forming repeats of STAND NTPases triggering the assembly of the multimeric complex in an NTP-dependent manner (*Danot et al., 2009*; *Hofmann, 2019*): for instance, in the classic AP-ATPase-based APAF-1 apoptosome, the Cytochrome c released from the mitochondria is sensed by the C-terminal WD40 repeats, which leads to the step-wise assembly of the heptameric apoptosome with the utilization of ATP/dATP by the STAND domain (*Figure 8B*). This in turn serves as a platform for the assembly of a multimeric caspase proteolytic cascade resulting in apoptosis (*Leipe et al., 2004*; *Chaudhary et al., 1998*; *Dorstyn et al., 2018*; *Zou et al., 1999*). Broadly similar assemblies of AP-ATPases and NACHT-NTPases are also central to the pathogen-triggered hypersensitivity response in plants and pyroptosis/apoptosis in animals (*Hofmann, 2019*; *Jones et al., 2016*). In the fungal hetero-incompatibility system, NACHT-NTPase proteins with similar architectures trigger programmed cell death when incompatible hyphal types fuse and result in interactions of the variable Het regions of these proteins (*Glass and Dementhon, 2006*). In addition to the formation of toroidal structures by the STAND NTPase proteins, homotypic interactions between the CARD domains of the Death-like superfamily also play a key role in in apoptosome-assembly (*Aravind et al., 1999*; *Wilson et al., 2009*). Similar interactions are also central to the vertebrate inflammasome network, where the NACHT-family STAND NTPase NLRP3 and the Pyrin domain protein AIM2 trigger oligomerization of the Death-like superfamily CARD and Pyrin domain proteins Mav3 and ASC to result in a prion-like multimeric autoassembly of the ASC protein by Pyrin-Pyrin interactions, which finally induces apoptosis (*Cai et al., 2014*; *Masumoto et al., 1999*; *Figure 8B*).

In the systems under consideration similar NTP-dependent multimeric assemblies are likely generated by the STAND NTPases and their various analogs (e.g. the tubulin and HSP70 system coupled to the DO-GTPases) while the interactions of the Death-like domains are likely paralleled by the structurally comparable EADs and the prokaryotic homologs of the Death-like superfamily (*Figure 8B–C*). Taken together, these aspects of the effectors from such systems imply that, whereas some are likely to act in a manner comparable to 'conventional' conflict systems by directly targeting invasive nucleic acids or proteins, the rest are more likely to function in less-commonly encountered mechanisms that involve: (i) decoy interactions to prevent associations between invasive molecules and host proteins, and (ii) formation of diverse macromolecular assemblages that might result in cell-death of the host or containment of the invasive entity (*Figure 8B*).

## Evolutionary considerations

The 4 vWA-MoxR-centric systems, 1 of the two DO-GTPase-centric systems, and the NucA- and EACC1- centric systems (7 of the eight above-reported systems) show a significant association with organisms displaying a multicellular or colonial habit or those with differentiated cell-types in course of their development (*Lyons and Kolter, 2015*; *Kysela et al., 2016*; *Figure 2*, *Figure 2—figure supplement 1*, *Table 1*). The vast majority of these organisms are bacteria belonging to evolutionarily

diverse clades (*Figure 1—source data 1*). These include: (1) Bacteria showing vegetative hyphae with multinucleate compartments and aerial hyphae that spawn sporogenic structures (e.g. *Streptomyces* [*Barka et al., 2016*]). (2) Bacteria forming filaments of multiple cells that might further be enclosed in a gelatinous sheet (e.g. *Nostoc* and *Beggiatoa*) (*Becerra-Absalón and Tavera, 2009*). Further, cyanobacteria show cellular differentiation with up to four distinct cell types in the most complex forms: vegetative photosynthetic cells, hormogonia, which are small motile dispersal filaments, 'stem-cell'-like akinetes and nitrogen-fixing non-reproductive heterocysts (*Dawkins and Krebs, 1979*; *Zhang et al., 2014*; *Flores, 2012*). (3) Organisms with complex biofilms, sociality, rosette formation (planctomycetes), colony-formation or structure-dependent cell differentiation (*Azospirillum* and *Archangium*) (*Rodrigues et al., 2015*; *Muñoz-Dorado et al., 2016*; *Strohl and Larkin, 1978*; *Wielgoss et al., 2019*; *Fuerst, 1995*). Notably, the few occurrences in archaea are also in those with known multicellularity (e.g. *Methanosarcina mazei*) (*Mayerhofer et al., 1992*) or a tendency to form tight cell-clusters (e.g. *Cuniculiplasma divulgatum* [*Golyshina et al., 2016*]). This strong association with multicellularity and sociality suggests that the distinctness of these systems is likely to be closely related to this organizational aspect of these organisms. Indeed, organisms with copies of more than one system 'type' are nearly-exclusively tied to multicellularity (*Figure 1—source data 1*). Moreover, some of these organisms, like actinobacteria, are slow-growers but are strong competitors in their environment. Thus, more generally it may be said that many of the prokaryotes with these systems have a more K-selected as opposed to r-selected life history (*Smith, 1998*; *van Elsas et al., 2007*). The sporadic presence of these systems in the core genome of evolutionarily distant lineages, which is more correlated with multicellularity rather than phylogeny (*Figure 2*), suggests that they likely originated in such lineages followed by extensive dissemination by lateral transfer. The presence of multiple paralogs of the ternary and GTPase-centric systems in the same organism, which are not closely related points to accretion of multiple copies of such through lateral transfer with each likely directed at a different set of invasive entities.

Further, while comprised of very disparate components, the above-noted thematic convergence is rather strong across these systems. This suggests that they have been repeatedly convergently shaped by comparable selective pressures co-evolving with the emergence of the multicellular state. Emergence of the multicellular or colonial states is predicated by assembly of clusters of cells which are kin (*Bourke, 2014*; *West et al., 2006*). The invasion of a single cell in such a cluster by an infective invasive entity like a virus presents considerable risk for its spread to the rest of the cells. Being kin, the cells have the benefit of inclusive fitness accrued via other cells in the assemblage (*Bourke, 2014*; *Hamilton, 1964*). Accordingly, when a cell is infected by the invader it pays to 'sacrifice itself' for the kin because such an act could limit the infection and still allow the cell to have net positive fitness via its kin (*Bourke, 2014*; *Hamilton, 1964*). Thus, we would expect apoptotic mechanisms coupled with immunity to emerge along with multicellularity (*Lyons and Kolter, 2015*; *Alteri et al., 2013*). However, since suicide could have fitness-depressing consequences if inappropriately triggered, we would expect that these systems are tightly regulated by threshold-sensing mechanisms which respond to the strength of infection rather than the mere detection of infection. This factor, together with the more K-selected life-histories of the organisms with such systems, are the likely selective forces driving thematic convergence among these systems.

A part of this thematic convergence comes from actual sharing of effector domains. We have previously reported extensive sharing of such domains between several distinct types of biological conflict systems (*Iyer et al., 2011a*; *Zhang et al., 2012*; *Zhang et al., 2016*; *Makarova et al., 2019*). Although these systems share some nucleotide-utilizing enzymatic domains and nucleic-acid-targeting enzymatic domains with several other conflict systems (*Burroughs et al., 2015*; *Brzozowski et al., 2019*), they generally have their own unique types of effector domains. However, several of these distinctive effector domains, EADs and signaling peptidases and nucleotide-utilizing enzymes are shared between the different types of systems reported in this study – this implies that they constitute their own effector-sharing evolutionary network which captures effectors from each other. However, their core components like the MoxR-vWA pair, DO-GTPase-GAP1 pair or the EACC1-caspase pair are not exchanged and distinguish these systems. Thus, while the MoxR-vWA pair appears to have had its origins in the more widespread CoxD-CoxE systems of bacteria (*Pelzmann et al., 2009*; *Pelzmann et al., 2014*), the DO-GTPase-GAP1 or the EACC1-caspase pair have no close relatives elsewhere. The presence of four distinct MoxR-vWA systems suggest that the ancestral CoxD-CoxE system was first exapted as a threshold-setting regulatory switch for pre-

existing conflict systems. These then rapidly diversified by the addition of a critical third component such as VMAP or iSTAND (*Figures 3B* and *5A*), which is likely to coordinate the sensing of the invasive entity and further processing of the effectors with the activation of the switch. This resulted in the formation of a ternary core, which then retained its distinctness with only the other remaining variable modules rapidly diversifying and being exchanged between parallel systems.

There are diverse origins for the effector modules in these systems. At least two of them appear to have reused sub-systems that occur independently in other conflict systems. First, the FtsH-ternary systems have used a remarkable complete tri-ligase Ubl-system (*Burroughs et al., 2011*; *Iyer et al., 2006*). Here, the EAD1 is fused to the Ubl and likely recruits the effectors to proteins to which the Ubl is conjugated via the E1, E2 and E3 ligase components of the system (*Figure 5B*). While some of these effectors, such as the caspase-like peptidase, might directly cleave target proteins, the Ubl conjugation might also route it for proteasomal degradation. This Ubl with the fused EAD1 is most likely cleaved off from the larger polypeptide, which it is part of, by the associated JAB-peptidase domain. Ubl systems with different Ubl-ligase complements are also part of other prokaryotic conflict systems, such as a version coupled to the nucleotide-activated effectors in the SMODS systems (*Burroughs et al., 2015*). We also found a Ubl-conjugation system related to those in the FtsH-ternary systems coupled with a Pup-conjugation system in certain bacteria of the *Thermofonsia* and Planctomycete lineage, lending support to the idea the conjugation of these Ubls might be linked to target-degradation (*Burroughs et al., 2015*). In eukaryotes, too, Ub modifications of effectors and signaling components have been incorporated into different immune and apoptotic responses (*Witt and Vucic, 2017*; *Tokunaga and Iwai, 2012*). Second, the previously-described DGR system for diversifying the FGS domains, which are present in several phages and bacteria, has also been incorporated into these systems (*McMahon et al., 2005*; *Le Coq and Ghosh, 2011*; *Figures 3B* and *4A,D*). They are the primary effectors of the VMAP-ternary system in cyanobacteria.

Actinobacteria parallel the cyanobacteria in showing great diversity in the effector modules of the VMAP-ternary systems even in closely-related species (*Figure 3B*). However, this occurs via the acquisition of a diverse panoply of effector domains rather than the diversification of a single domain through a mutagenic system. Examination of their genomic neighborhoods revealed no association with a DGR element or any other potential mutator element. However, we noticed a statistically significant preferred location for VMAP-ternary systems in the genome at approximately 33% of the length of the chromosome from either end in the linear chromosomes of Streptomycetales (n = 141 distinct organisms; $\chi$-squared test p=1.628e-07 for null hypothesis of uniform as opposed to bimodally clustered distribution across 20 chromosomal intervals). These two symmetric preferred locations (*Figure 8D*) bound the chromosomal 'core', which is enriched in the conserved and housekeeping genes (*Hopwood, 2006*). As opposed to the core, the arms which lie close to the ends of the chromosome show an enrichment of genes for specialized secondary metabolism and conflict systems as well as transposons and their remnants. The arms are also hotspots of numerous non-homologous end-joining-associated rearrangements such as deletions, circularization and arm exchange (*Hoff et al., 2018*). Accordingly, we posit that the preferred position of the VMAP-ternary system (*Figure 8D*), especially in actinobacteria with large linear chromosomes, allows them to undergo frequent variations by a recombinational mechanism that affects the chromosome arms. Additionally, the multinucleate nature of hyphae of such actinobacteria (*Hopwood, 2006*) probably allows recombination between different chromosomes to foster variability of the VMAP-ternary systems.

Finally, it is notable that these threshold-regulated systems of prokaryotes find close parallels in apoptotic systems of multicellular eukaryotes, which are triggered by physical interactions with proteins of invasive entities or apoptotic stimuli (*Park et al., 2007*; *Aravind et al., 1999*; *Wilson et al., 2009*). In terms of direct evolutionary connections, we expand the set of domains that are common to both repertoires to now include the Death-like fold and the TRADD-N domains. This reinforces the earlier proposal that eukaryotic apoptotic systems emerged from component domains acquired via lateral transfer from prokaryotes (*Koonin and Aravind, 2002*; *Aravind et al., 2001*; *Aravind et al., 1999*; *Hofmann, 2019*). However, it also points to a more subtle feature: in both the eukaryotic and prokaryotic cases, comparable sets of domains appear to have organized themselves into systems based on common principles, which combine invader-sensing with threshold-setting for activation of terminal effectors through multimerization, nucleotide-signals and proteolytic cascades.

These are most obviously seen in the convergent domain architectures, such as those of the EADs and the DEATH-like domains, especially in terms of their coupling with various effector components (*Figure 8A*). This implies that in addition to the basic 'vocabulary' in the form of several shared domains there is also a shared 'grammar' of the similar architectures and likely interaction networks of these domains across these systems. This in turn suggests that the spread of a relatively small set of protein domains along with the repeated emergence of this 'grammar' in their organization has gone hand-in-hand with the multiple emergences of multicellularity across the tree of Life.

## Conclusions

The above-described systems greatly add to the diversity of themes and effectors observed in biological conflicts. The complex thresholding systems that we reconstruct for the MoxR-vWA and DO-GTPase systems have little precedent in the described universe of biological conflict systems. Based on relatively limited genomic data, we had formerly noted that several components typical of animal, fungal and plant apoptotic systems had their origins in bacteria and are enriched in taxa with a multicellular organization (*Koonin and Aravind, 2002*; *Aravind et al., 2001*; *Aravind et al., 1999*; *Hofmann, 2019*). The results in this article indicate that this association holds with a much more phyletically extensive dataset and expands to include: (1) multiple highly regulated systems with similar organizational principles; (2) additional domains uniquely shared with animal apoptotic systems (e.g. TRADD-N and Death-like domains); (3) common principles regarding the coupling of immune responses against invasive entities with programmed cell-death, which is also an important feature of animal and plant immunity (*Jones et al., 2016*); (4) Effectors acting through formation of multimeric complexes and superstructures.

Accordingly, we propose that the selection for multiple such systems went alongside the emergence of multicellularity and differentiation and that their appropriate regulation is of general importance for the stability of such cellular organizational states. In line with such a proposal, we observed that in these systems there are often two levels of regulation as their faulty deployment could have negative consequences. We propose that the vWA-MoxR pair or the GTPase-GAP1 pair represents the first such level of such regulation, which is coupled to direct sensing of the invasive entity (e.g. by the VMAP-M domains of the VMAP protein or the FNIII domains of GAP1-N2) (*Figures 3B*, *5A–C* and *6B*). However, it appears that in many of these systems a second threshold is set in the form of the activation of the associated signaling domains: e.g. the peptidases and the nucleotide-utilizing enzymes. This finally releases the effectors by proteolysis or activates them through a nucleotide-derived signal. Further, in a multicellular assemblage there is much greater need for signaling the presence of the infection and limiting its spread to other cells (*Bourke, 2014*; *Hamilton, 1964*). This explains features unique to these systems: (1) the presence of coupled systems involved in production of a processed and/or modified peptide which could act as a signal to alert non-infected cells of the presence of an impending infection thereby triggering other immune mechanisms in them (*Figures 3A–B* and *5B*). (2) The super-structure-forming and multimerizing effectors and EADs, or 'neutralizing' effectors like the FGS domains, which could physically interact with the invasive entity and block its spread. Likewise, the effectors with inactive versions of various enzymatic domains could act as decoys to interact with the invasive entities and block their interactions with the active versions that might be their typical targets. (3) Finally, the presence of several membrane-associated versions among these systems implies that they are geared for the interception of the invasive entities even as they enter the cell. This kind of surveillance would again be useful in protecting the multiplicity of cells that typify a multicellular or social system.

We hope that this study inspires further biochemical and ecological investigation of these systems to study the precise stimuli that activate them. We also believe that these systems have potential for biotechnological applications – the diversifying FGS and VMAP-M domains could be used as analogs of antibodies, whereas the dominant negative enzymatic domains could be used to probe cellular functions both in these multicellular bacteria and conventional model systems.

## Materials and methods

### Sequence analysis

Iterative sequence profile searches were performed using the PSI-BLAST (RRID:SCR_001010) and JACKHMMER programs. Similarity-based clustering for both classification and culling of nearly identical sequences was performed using the BLASTCLUST program (ftp://ftp.ncbi.nih.gov/blast/documents/blastclust.html) (RRID:SCR_016641). The length (L) and score (S) threshold parameters were variably adjusted as needed. For example, the length (L) and score (S) threshold parameters for clustering near identical proteins was L = 0.9 and S = 1.89. HMM searches were run using either HMMsearch initiated with a HMM built from an alignment or iteratively using JACKHMMER from a single sequence. The sequence databases against which these searches were run were: (1) the non-redundant (*nr*) protein database frozen at June 21 2019 of the National Center for Biotechnology Information (NCBI) (*Broderick et al., 2014*; *Barnett et al., 2000*); (2) this *nr* clustered down to 50% similarity using the MMseqs program (*Hauser et al., 2016*) (RRID:SCR_008184); (3) A custom database of 7423 complete prokaryotic genomes extracted from the NCBI Refseq database. The HHpred program (RRID:SCR_010276) was used for profile-profile searches (*Zimmermann et al., 2018*) and run against (1) HMMs derived from PDB; (2) Pfam models; (3) A custom database of alignments of diverse domains curated by our group. For previously known domains, the Pfam database (*Finn et al., 2016*) was used as a guide, although the profiles were corrected for their boundaries where required and augmented by addition of newly detected divergent members that were not detected by the original Pfam models. All novel alignments emerging from this study can be accessed in the Supplemental material. Multiple sequence alignments were built by the Kalign (*Lassmann et al., 2009*) (RRID:SCR_011810) and PCMA (*Pei et al., 2003*) programs followed by manual adjustments on the basis of profile-profile and structural alignments. Secondary structures were predicted using the JPred program (*Cole et al., 2008*) (RRID:SCR_016504).

### Other procedures

Structure similarity searches were performed using the DaliLite program (*Holm, 2019*; *Holm et al., 2008*) (RRID:SCR_003047). Structural visualization and manipulations were performed using the PyMol (http://www.pymol.org) program (RRID:SCR_000305). Contextual information from prokaryotic gene neighborhoods was retrieved using a Perl script that extracts the upstream and downstream genes of the query gene from the GenBank genome file and uses BLASTCLUST to cluster the proteins to identify conserved gene-neighborhoods. Recognition of gene neighborhood conservation relied on several filters including: (1) nucleotide distance constraints (generally 70 nucleotides); (2) conservation of gene directionality within the neighborhood; (3) presence in more than one phylum. Phylogenetic analysis was conducted using an approximately maximum-likelihood method implemented in the FastTree 2.1 (RRID:SCR_015501) program under default parameters (*Price et al., 2010*). Perl scripts were used to run automated large-scale analysis of sequences, structures and genome context as described above. Network analysis (igraph and circlize [*Gu et al., 2014*] packages), data processing (knitr (*Xie, 2014*) and dplyr packages), analysis and visualization was performed using the R language. Position-wise Shannon entropy values for domains were calculated as described in *Krishnan et al. (2018)*.

### Statistical tests for significance

Tests for significance of association of systems with multicellularity were performed thus: (1) Each organism in the above-mentioned curated prokaryotic genome database assembled from NCBI Genbank/RefSeq was systematically assessed and assigned a multicellularity flag (True, False, Not Applicable) using all available information obtained from the Bergey's Manual of Systematic Bacteriology (*Whitman, 2015*), and the latest publications on the individual taxon. This was done for the 6956 organisms in the database (*Supplementary file 1*), which accounts for all the prokaryotic organisms in it other than the candidate phyla radiation (CPR) for which no information exists. (2) This gave us a set background frequency of organisms with or without a multicellular habit which we could use to test the significance of the associations of the systems reported here with multicellularity using the hypergeometric distribution implemented in the *phyper* command of the R language. For this test, the four input values were: q = the number of organisms containing a copy of a given system which

score as multicellular; m = the total number of multicellular organisms in the database; n = total number of non-multicellular organisms in the database; k = total number of organisms with the given system [drawn without replacement from the total set in the database].

The $\chi^2$ test for the preferential location of VMAP-ternary systems in the genome at approximately 33% was performed using the implementation in *chisq.test* command in the R language (see above for details).

# Acknowledgements

This research was supported by the Intramural Research Program of the NIH, National Library of Medicine.

# Additional information

### Funding

| Funder | Grant reference number | Author |
|---|---|---|
| National Institutes of Health | Intramural Research Program | Gurmeet Kaur<br>A Maxwell Burroughs<br>Lakshminarayan M Iyer<br>L Aravind |

The funders had no role in study design, data collection and interpretation, or the decision to submit the work for publication.

### Author contributions

Gurmeet Kaur, Formal analysis, Investigation, Visualization, Methodology; A Maxwell Burroughs, Conceptualization, Formal analysis, Supervision, Investigation, Visualization, Methodology; Lakshminarayan M Iyer, Formal analysis, Validation, Investigation, Visualization; L Aravind, Conceptualization, Data curation, Formal analysis, Supervision, Funding acquisition, Validation, Investigation, Visualization, Methodology, Project administration

### Author ORCIDs

A Maxwell Burroughs (iD) https://orcid.org/0000-0002-2229-8771

L Aravind (iD) https://orcid.org/0000-0003-0771-253X

### Decision letter and Author response

Decision letter https://doi.org/10.7554/eLife.52696.sa1

Author response https://doi.org/10.7554/eLife.52696.sa2

# Additional files

### Supplementary files

• Supplementary file 1. List of multicellular status of all organisms included in analysis of significance of systems overrepresentation in multicellular prokaryotic genomes. Multicellular flag is defined as either TRUE, FALSE, or NA (not enough information available) (see Materials and methods).

• Transparent reporting form

### Data availability

All data generated or analysed during this study are included in the manuscript and supporting files.

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
