## [Decision Letter]

**Acceptance summary:**

The paper is based on an in-depth computational analysis of genomic information that is used to interpret lifestyles in terms of the influence of three different classes of chaperone-based protein modules. The study describes a biological scenario that links a molecular system with general evolutionary properties, in the key area of the emergence of multicellularity.

**Decision letter after peer review:**

Thank you for submitting your article "Highly-regulated, diversifying NTP-based biological conflict systems with implications for emergence of multicellularity" for consideration by *eLife*. Your article has been reviewed by Aleksandra Walczak as the Senior Editor, a Reviewing Editor, and three reviewers. The following individuals involved in review of your submission have agreed to reveal their identity: Laszlo Nagy (Reviewer #3).

The reviewers have discussed the reviews with one another and the Reviewing Editor has drafted this decision to help you prepare a revised submission.

Summary:

In this paper the analysis of genomes at different levels is used to propose a model that relates three classes of chaperone-based modules: (a) chaperone/co-chaperone-based systems of MoxR-like AAA+ ATPase and its co-chaperone von Willebrand factor A (vWA) domains, (b) GTPases centered systems and (c) those dependent on proteolytic cascades) with multicellular lifestyles in bacteria, suggesting the relation with eukaryotes systems and possibly with the emergence of multicellularity.

Essential revisions:

The two related key points for the revision are:

The paper has to show explicitly the association of the presence and copy numbers of the analyzed systems with the presence/absence of a multicellular stage in the life cycle across the taxa analyzed. Additionally, it has to demonstrate the statistical reliability of the proposed association between the genetic systems and multicellularity.

In practical terms, the manuscript is not well organized and difficult to follow (both page numbers and line numbers are missing). It has to be reorganized substantially making clear the differences between the different hypotheses/results (for example, one of the reviewers suggests that the various subgroups of MoxR-like systems could be more concise and much of it could be moved to supplementary information). Importantly, the reorganization should not be done at the expenses of reducing the contend, given that the specific findings have actual value by themselves (e.g. novel VMAP-C domain that could function as a potential protein interaction-mediating module).

Additionally, the paper is poorly referenced. There are statements (or even paragraphs) that clearly sound like literature facts, but references are not given in many cases, making it hard to understand what is novel and what is not.

1) "The most confusing aspect of this manuscript to me was the initial identification of these domains described in the first paragraph of the Results section. For example, I don't understand how the initial database of conflict systems was generated. I also don't understand the significance of the sentence starting with "We then used the fact that…" and how variability was used in this analysis. Due to this lack of understanding, when I read the manuscript, I was not sold on the idea that these systems were really involved in biological conflict. […] I think the paper would be strengthened by doing a better job expanding this initial paragraph to really convince the reader that these systems have a potential in biological conflict."

2) One point that was not discussed is that AAA+ ATPase domains have been demonstrated to be regulated by c-di-GMP (and by extension perhaps other c-di-nucleotides). For example, the transcription factors FleQ in *P. aeruginosa*, FlrA, and VpsR in *V. cholerae* along with AAA+ involved in Type II Secretion Systems are regulated directly by c-di-GMP. Is it worth speculating that cdNs generated by SMODS, which are also involved in biological conflict, could cross talk with the chaperones via this domain? Are SMODS ever co-localized with these systems?

3) In the "two-component conflict system" section the authors reference Figure 5C, but I think they mean Figure 6C. As this system is not a typically two-component system (i.e. histidine kinase and response regulator), I would avoid using this term as it will generate confusion. Maybe use "two-part" or something analogous.

4) In the evolutionary considerations section, the authors discuss the ubiquitin-like systems. It is worth noting that Sorek's recent Nature paper demonstrated a role for these in phage defense using a genetic approach.

5) How similar are the inferred peptide residues of MoxR associated vWA with that of classical vWA peptide-binding residues? Are there any compensatory mutations in those vWA that lack one or two acidic residues that could bind Mg^2+^.

6) It is not clear if any particular α-proteobacteria has all three major regulatory systems?

7) Figure 3A: The network figure is a bit dense and makes it difficult to understand the details, as there are many edges. Can the authors aim to connect groups (nodes in the same color) with only one edge?

8) Can the thickness of edges be modified in the network Figure 4A? i.e. directed edges from different iSTAND types to MoxR then to effectors could be appropriately represented (in terms of thickness) to reflect the number of instances such systems are observed in genomes/metagenomes.

9) Figure 7 could be a summary/model figure highlighting the central findings of the manuscript. Figure 7A could be represented in the alignment (Figure 7B) and histograms (Figure 7C) could be moved to supplementary information.

10) Horizontal transfer the protein modules are found on horizontally transferred genetic elements or part of core genomes?

11) Did the authors require the three required genes of the ternary system to form a gene cluster or operon? If yes, what was the operational definition of a gene cluster?

12) Subsection “The MoxR-vWA-centric ternary systems” – there are at least 4-5 different systems that the authors deal with in this manuscript. A schematic figure of the properties of these (e.g. combinations of genes/domains) would be useful. This would also come handy when discussing the four subtypes of the ternary systems.

13) Subsection “The MoxR-vWA-centric ternary systems” – "current *nr* database " – you mean the NCBI database or your own assembled database of prokaryotic genomes?

14) Subsection “The distinct features of the core components of the VMAP-ternary systems” – this should be shown on a schematic tree figure preferably with the multicelllar stages of the species shown.

15) *Subsection “Evolutionary considerations”* – "8 of the 9 above-reported systems are solely or primarily found in organisms with a multicellular or colonial habit or those with differentiated cell-types in course of their development. " this should be explicitly demonstrated by statistical analyses – as mentioned above as a key point for the revision.

16) *Subsection “Evolutionary considerations”* – "repeatedly convergently shaped by strong selective pressures" too colloquial. Why would selection pressures be strong? This paragraph discusses common theories of multicellularity, but no references are provided.

---

## [Author Response]

Essential revisions:The paper has to show explicitly the association of the presence and copy numbers of the analyzed systems with the presence/absence of a multicellular stage in the life cycle across the taxa analyzed.

We have addressed this concern by leveraging the curated prokaryotic genome database we had assembled from NCBI Genbank/RefSeq described in the Materials and methods section: we systematically assessed each organism in the database and assigned it a multicellularity flag (True, False, Not Applicable) using all available information obtained from the Bergey’s Manual of Systematic Bacteriology, and the latest publications on the individual taxon. This was done for the 6956 organisms in the database (Supplementary file 1), which accounts for all the prokaryotic organisms in it other than the candidate phyla radiation (CPR) for which no information exists. All representatives of each type of system are now provided as an itemized list in the Figure 1—source data 1 file which contains organism name, system component composition, and multicellularity flag (TRUE, FALSE, NA). To visually underscore the link between these systems and multicellularity, this information has also been summarized in the figures (Figure 2, Figure 2—figure supplement 1) added in response to one of the below comments.

Additionally, it has to demonstrate the statistical reliability of the proposed association between the genetic systems and multicellularity.

We used the above assignment to test each system for the association with multicellularity by chance alone using the hypergeometric distribution (see new subsection added to Material and Methods section):

p-hypergeometric (# of organisms containing a copy of a given system which score as multicellular, total # of multicellular organisms in the database, total # of non-multicellular organisms in the database, total # of organisms with the system [without replacement]).

We have added the details of this test to the Materials and methods section. The results of this test are illustrated in Figure 2—figure supplement 1 and provided it in Table 1 of the manuscript with the underlying data in Figure 1—source data 1. This analysis supports our proposal that the systems (with the earlier noted exception of the GAP1-N1 systems) correlate strongly with a multicellular habit in prokaryotes.

In practical terms, the manuscript is not well organized and difficult to follow (both page numbers and line numbers are missing). It has to be reorganized substantially making clear the differences between the different hypotheses/results.

We apologize for the oversight and have added page numbers and line numbers. As to the second point, we have put some effort into planning the organization of the paper, to properly introduce the many complexities of these systems that are required to reach the conclusions that have been drawn. To the extent possible, we have isolated the description of the findings (results) in the initial sections from the interpretations of these results (hypothesis/ discussion) in the final sections. To better emphasize this, we have also reorganized the sub-levels to allow for better navigations through the different types of systems. Further, we have added a new summary subfigure (Figure 8A) that helps a reader obtain an overview of the systems at a glance focus particular ones of interest.

(For example, one of the reviewers suggests that the various subgroups of MoxR-like systems could be more concise and much of it could be moved to supplementary information). Importantly, the reorganization should not be done at the expenses of reducing the contend, given that the specific findings have actual value by themselves (e.g. novel VMAP-C domain that could function as a potential protein interaction-mediating module).

We have attempted to streamline individual sections without removing content, as requested. However, we retain the sub-groups of the MoxR-like systems because each of them shows distinct features that are essential to build the overall case for these forming a unified theme of biological conflict systems.*Additionally, the paper is poorly referenced. There are statements (or even paragraphs) that clearly sound like literature facts, but references are not given in many cases, making it hard to understand what is novel and what is not.*

In balancing the need to introduce a wide range of concepts with the need to be concise, extended Discussion section with accompanying references in certain places where we felt the material was well-accepted were not originally included. Based on the reviewer’s suggestion, we have re-evaluated these places and added new references while keeping discussion of the concepts to a minimum, staying in line with the previous comment. In some places in the text where no references are given the observation is being reported, to our knowledge, for the first time. We try to clarify these instances.*1) "The most confusing aspect of this manuscript to me was the initial identification of these domains described in the first paragraph of the Results section. For example, I don't understand how the initial database of conflict systems was generated.*

Identification of the novel conflict systems introduced in this manuscript was in part dependent on the discovery of shared domains and organizational principles between previously characterized conflict systems and these novel systems. To clarify this point, we have emended the paragraph in question and added references which better lay out our methodological choices (subsection “The MoxR and vWA components”).

I also don't understand the significance of the sentence starting with "We then used the fact that…" and how variability was used in this analysis. Due to this lack of understanding, when I read the manuscript, I was not sold on the idea that these systems were really involved in biological conflict.

Of the genomic indicators of biological conflict systems, among the strongest that have been documented in the past are: (1) high variability in domain architecture via displacement of particular domains, while retaining the same overall architectural “grammar” typical of multiple biological conflict systems. (2) variability with regards to rapid sequence divergence within in domains. (3) High degree of lateral transfer and difference in presence and absence between closely related organisms. The references for these observations have been provided, but we have better explained the intent behind the use of the word “variability” in the and expanded it with the further explanation in response to this comment of the reviewer to clarify our reasoning (subsection “The MoxR and vWA components”). We also emphasize the detection of these features in the systems under consideration (e.g. subsection “vWA-MoxR Subsystem 1: the variable effector domains of the VMAP-ternary systems”). After laying out all the components of the systems we identified, we summarize our arguments for why these should be biological conflict systems in the Conclusion section. Therein we consider the totality of features that are comparable to other conflict systems and sum up our inferences regarding the proposed conflict functions.

I think the paper would be strengthened by doing a better job expanding this initial paragraph to really convince the reader that these systems have a potential in biological conflict.

Please see the above two responses. We have accordingly modified the introductory paragraph of the Results and Discussion section laying out the assumptions we leveraged to identifying these novel systems.

2) One point that was not discussed is that AAA+ ATPase domains have been demonstrated to be regulated by c-di-GMP (and by extension perhaps other c-di-nucleotides). For example, the transcription factors FleQ in P. aeruginosa, FlrA, and VpsR in V. cholerae along with AAA+ involved in Type II Secretion Systems are regulated directly by c-di-GMP. Is it worth speculating that cdNs generated by SMODS, which are also involved in biological conflict, could cross talk with the chaperones via this domain? Are SMODS ever co-localized with these systems?

The reviewer raises a point of interest. In our analyses we did not observed co-localization of these genes mentioned by the referee with the SMODS-based systems (or any individual components thereof). Nor did we identify association with a comparable nucleotide-generating enzyme which could potentially add a further layer of thresholding regulation on these systems. FleQ, FlrA and VpsR are σ-54 dependent AAA+ ATPase (NtrC-like) transcription factors. We had previously proposed that one such (RtcR) recognizes cyclic nucleotide; however, the nucleotide-binding domain found in that protein is absent in FleQ, FlrA and VpsR. But as ligand activated-transcription factors these NtrC-like proteins could recognize activating nucleotide ligands through other mechanisms. We have added this reference to the revised manuscript (Results and Discussion section).

3) In the "two-component conflict system" section the authors reference Figure 5C, but I think they mean Figure 6C.

The reviewer is correct, we have made this change to the text.

As this system is not a typically two-component system (i.e. histidine kinase and response regulator), I would avoid using this term as it will generate confusion. Maybe use "two-part" or something analogous.

We have changed "two-component conflict system" to read “two-gene conflict system”.

4) In the evolutionary considerations section, the authors discuss the ubiquitin-like systems. It is worth noting that Sorek's recent Nature paper demonstrated a role for these in phage defense using a genetic approach.

The role of ubiquitin-like systems in conflict has been described at length, independent and well prior to the recent paper mentioned by the reviewer in multiple publications. We have already cited the original papers relevant to this discussion.

5) How similar are the inferred peptide residues of MoxR associated vWA with that of classical vWA peptide-binding residues? Are there any compensatory mutations in those vWA that lack one or two acidic residues that could bind Mg2+.

The inferred peptide-binding residues of the ternary systems vWA domain family are positionally-equivalent and identical to the classical vWA peptide-binding residues. We can hence say with confidence that the vWA domains lacking one or both acidic residues clearly lack the equivalent residues. They do not show any obvious compensatory acidic residues but have conservation of other residues in the equivalent positions which might imply an alternative mechanism of binding via those residues. However, these are a small set relative to those which retain the conserved complement of peptide-binding residues.

To clarify the above points, we have done the following: (1) we have split the ternary system vWA alignment in the Figure 1—source data 1 to render it into a more readable form; (2) we have flagged the positions of the conserved aspartate residues in this alignment; (3) we have also added positional flags to the alignment of the smaller vWA family to highlight the missing aspartate residues; and (4) we have added a specific pointer to these alignments in the text.

6) It is not clear if any particular α-proteobacteria has all three major regulatory systems?

While no single α-proteobacterium, or any other bacterium for that matter, is observed with all major regulatory system types. Newly provided source data (Figure 1—source data 1) have been added to include an overview of the overlap between the systems and lists of the organisms containing more than one system. We have added pointers to this source data in the manuscript in the evolutionary considerations section. There is a notable general overlap between the VMAP ternary, GAP1-N2, and EACC1 systems (22-44% of organisms have at least 2 of these). Similarly, the overlap between the FtsH and β-propeller systems is approximately 17%. The rest show considerably lower overlap.

7) Figure 3A: The network figure is a bit dense and makes it difficult to understand the details, as there are many edges. Can the authors aim to connect groups (nodes in the same color) with only one edge?

On the surface this recommendation presents a solution to network overcrowding, yet there are several problems with its implementation. Most notably, while we do observe a general clustering of functional likeness in the network, there are several exceptions which would be difficult to account for by implementing this suggestion. More fundamentally, the network is meant to convey the functional complexity and sheer diversity of the links to the effectors we observe in the system and reducing these to a single node would seem to undermine that affect. However, we do include simplified network representations to illustrate the principles of linkage for each of the major themes (the ensuing sub-figures B-D now part of Figure 4). In response to the referee’s comment to make the network more readable we have grouped and labeled each group. These groupings correspond to the discussion in the text.

8) Can the thickness of edges be modified in the network Figure 4A? i.e. directed edges from different iSTAND types to MoxR then to effectors could be appropriately represented (in terms of thickness) to reflect the number of instances such systems are observed in genomes/metagenomes.

In the case of the iSTAND systems the different inputs to the MoxR node are roughly equivalent, to the extent that rendering any differences would border on the imperceptible. Given this, we have not implemented the suggestion. The raw data for the iSTAND systems is now provided in an easy-to-read itemized list as outlined in the above response to the essential revisions. Perusal of this table helps underscore the even distribution of these inputs.

9) Figure 7 could be a summary/model figure highlighting the central findings of the manuscript. Figure 7A could be represented in the alignment (Figure 7B) and histograms (Figure 7C) could be moved to supplementary information.

We have added a summary flow chart as recommended by the reviewer (Figure 8A). But we have kept the architectures and alignment separate for that explains the parallelism between the eukaryotic and prokaryotic architectures in a better way.

10) Horizontal transfer the protein modules are found on horizontally transferred genetic elements or part of core genomes?

The systems reported in this paper, with the exception of the diversity generating retrons which might show mobility as distinct elements, are all predicted to be a part of the core genome. The distribution patterns of their acquisition, particularly coupled with the observed shared lifestyle patterns, necessitate their acquisition via horizontal gene transfer. We have added language to the text in the subsection “Evolutionary considerations” which clarified their presence in the core genome.

11) Did the authors require the three required genes of the ternary system to form a gene cluster or operon?

Ternary system genes, and indeed, genes belonging to the other multi-gene systems described in the manuscript were all initially detected by requiring the three genes to be coupled in an operon in the genome. However, we expanded our search to detect those versions which were partial i.e. having two genes and also the standalone copies. The majority of the later (>65%) were from whole genome shotgun contigs making in unclear if they were merely fragmented due to assembly issues or truly isolated components. In cases where we were sure of genomic completeness the uncoupled components were nearly always found elsewhere on the genome.

If yes, what was the operational definition of a gene cluster?

The operational definition of the presence of a system in a given genome was the adjacent co-occurrence genes coding for the conserved components of the system in a common neighborhood preserved across at least two phylogenetically distinct prokaryotic lineages. The separation of the genes within such neighborhood was typically less than 25 nucleotides though for the actual detection we allowed a larger separation, which was experimentally demonstrated in the past as being usually up to 70 nucleotides with conservation of gene directionality (we specify these in the Materials and methods section of the revised manuscript).

*12) Subsection “The MoxR-vWA-centric ternary systems” – there are at least 4-5 different systems that the authors deal with in this manuscript. A schematic figure of the properties of these (e.g. combinations of genes/domains) would be useful. This would also come handy when discussing the four subtypes of the ternary systems*.

In accordance with this recommendation, we have added a final flow chart/model as Figure 8A as a reference point for the discussion of the unifying, shared features across the systems and the evolutionary observations made in the final sections. However, each system is also identified in very general terms in Figure 1A, we have added some additional pointers in the text to help the reader keep the distinctions between the different systems straight. Each distinct system is then further represented by simplified/idealized schematic depictions which detail their major components (see Figure 3B, Figure 4B-C, Figure 5A-C, and Figure 6A-B), which may be used with the summary figure.

13) Subsection “The MoxR-vWA-centric ternary systems” – "current nr database " – you mean the NCBI database or your own assembled database of prokaryotic genomes?

The explanation with regards to the databases used in our sequence searches was not entirely clear. The text flagged by the reviewer refers to the non-redundant (*nr*) database hosted at the NCBI. We have added a date for the last version of *nr* we used. We discuss *nr* in the Materials and methods section but inadvertently failed to mention subsection “Systematic identification of conflict systems with novel thresholding mechanisms”. We have added a line to the Results section which clarifies this, along with a pointer to the Materials and methods section where the different databases are laid out.*14) Subsection “The distinct features of the core components of the VMAP-ternary systems” – this should be shown on a schematic tree figure preferably with the multicelllar stages of the species shown.*

We illustrated this with a schematic tree as suggested (Figure 2).*15) Subsection “Evolutionary considerations” – "8 of the 9 above-reported systems are solely or primarily found in organisms with a multicellular or colonial habit or those with differentiated cell-types in course of their development. " this should be explicitly demonstrated by statistical analyses – as mentioned above as a key point for the revision.*

Please see the responses to the essential revisions above for the statistical tests we have performed and the data we have included to demonstrate this.*16) Subsection “Evolutionary considerations” – "repeatedly convergently shaped by strong selective pressures" too colloquial. Why would selection pressures be strong?*

We have modified this in the revised text. The arguments for the selective pressures favoring convergence organization have been elaborated in the prior paragraphs.

This paragraph discusses common theories of multicellularity, but no references are provided.

We have added references.